# The Role of Vadose Zone Physics in the Ecohydrological Response of a Tibetan Meadow to Freeze-Thaw Cycles

Lianyu Yu[1], Simone Fatichi[2], Yijian Zeng[1], Zhongbo Su[1,3]

[1] Faculty of Geo-information Science and Earth Observation (ITC), University of Twente, Enschede, The Netherlands

[2] Department of Civil and Environmental Engineering, National University of Singapore, Singapore

[3] Key Laboratory of Subsurface Hydrology and Ecological Effect in Arid Region of Ministry of Education, School of Water and Environment, Chang'an University, Xi'an, China

*Correspondence to*: Yijian Zeng (y.zeng@utwente.nl); Zhongbo Su (z.su@utwente.nl)

**Abstract.** The vadose zone is a sensitive zone to environmental changes and exerts a crucial control in ecosystem functioning, and even more so in cold regions considering the rapid change of seasonally frozen ground under climate warming. While the way in representing the underlying physical process of vadose zone differs among models, the effect of such differences on ecosystem functioning and its ecohydrological response to freeze-thaw cycles is seldom reported. Here, the detailed vadose zone process model STEMMUS was coupled with the ecohydrological model T&C to investigate the role of influential physical processes during freeze-thaw cycles. The physical representation is increased from using T&C without, and with explicit consideration of the impact of soil ice content on energy and water transfer properties, to T&C coupling with STEMMUS enabling the simultaneous mass and energy transfer in the soil system (liquid, vapor, ice). We tested model performance with the aid of a comprehensive observation dataset collected at a typical meadow ecosystem on the Tibetan Plateau. Results indicated that: i) explicitly considering the frozen soil process significantly improved the soil moisture/temperature profile simulations and facilitated our understanding of the water transfer processes within the soil-plant-atmosphere continuum; ii) the difference among various representations of vadose zone physics have an impact on the vegetation dynamics mainly at the beginning of the growing season; iii) models with different vadose zone physics can predict similar interannual vegetation dynamics, energy, water, and carbon exchanges at the land surface. This research highlights the important role of vadose zone physics for ecosystem functioning in cold regions and can support the development and application of future Earth system models.

## 1. Introduction

Recent climatic changes have accelerated the dynamics of frozen soils in cold regions, as for instance favoring permafrost thawing and degradation (Cheng and Wu, 2007; Hinzman et al., 2013; Peng et al., 2017; Yao et al., 2019; Zhao et al., 2019). In consequence of these changes, vegetation cover and phenology, land surface water and energy balances, subsurface soil hydrothermal regimes, water flow pathways were reported to be affected (Wang et al., 2012; Schuur et al., 2015; Walvoord and Kurylyk, 2016; Gao et al., 2018; Campbell and Laudon, 2019; Yu et al., 2020). Understanding how an ecosystem interacts with changing environmental conditions is a crucial yet challenging problem of Earth system research for high latitude/altitude regions and deserves further attention.

Land surface models, terrestrial biosphere models, ecohydrology models, and hydrological models have been widely utilized to enhance our knowledge in terms of land surface processes, ecohydrological processes (Fatichi and Ivanov, 2014; Fatichi et al., 2016a), and freezing/thawing process (Ekici et al., 2014; Wang et al., 2017; Cuntz and Haverd, 2018; Wang and Yang, 2018; Druel et al., 2019). By either incorporating a permafrost model into the ecosystem model (Zhuang et al., 2001; Wania et al., 2009; Lyu and Zhuang, 2018) or equipping the soil model with vegetation dynamics and carbon processes (Zhang et al., 2018), the temporal dynamics of soil temperature, permafrost dynamics and vegetation and carbon dynamics can be simultaneously simulated over cold region ecosystems. Moreover, the incorporation of detailed vadose zone and land surface processes (e.g., soil hydrology and snow cover) usually improves the model performance (Lyu and Zhuang, 2018) and facilitates model ability to investigate the ecosystem response to variations in climatic and environmental conditions at various spatial-temporal scales (Zhang et al., 2018). The importance of non-growing-season processes (e.g., freeze-thaw cycle, snow cover) was highlighted when interpreting the carbon budget observations and can significantly alter the carbon cycling and future projection of cold region ecosystems (Zhuang et al., 2001; Wania et al., 2009; Lyu and Zhuang, 2018; Zhang et al., 2018).

However, in most of the current modelling research in cold region ecosystems, the water and heat transfer process in the vadose zone remains independent and not fully coupled. Such consideration of vadose zone physics might result in unrealistic physical interpretations, especially for soil freezing/thawing processes (Hansson et al., 2004). In this regard, researchers have stressed the necessity to simultaneously couple the water and heat transfer process in dry/cold seasons (Scanlon and Milly, 1994; Bittelli et al., 2008; Zeng et al., 2009a; Zeng et al., 2009b; Yu et al., 2016; Yu et al., 2018). Concurrently, researchers developed dedicated models, e.g., SHAW (Flerchinger and Saxton, 1989), HYDRUS (Hansson et al., 2004), MarsFlo (Painter, 2011) and its successor Advanced Terrestrial Simulator (Painter et al., 2016), and STEMMUS-FT (Yu et al., 2018; Yu et al., 2020), implementing the soil water and heat coupling physics for frozen soils (see for reviews of the relevant models in Kurylyk and Watanabe, 2013; Grenier et al., 2018; Lamontagne-Halle et al., 2020). Promising simulation results have been reported for the soil hydrothermal regimes. While these efforts mainly focus on understanding the surface and subsurface soil water and heat transfer process (Yu et al., 2018; Yu et al., 2020) and stress the role of physical representation of freezing/thawing process (Boone et al., 2000;

Wang et al., 2017; Zheng et al., 2017), they rarely take into account the interaction with vegetation and carbon dynamics.

With the largest area of high-altitude permafrost and seasonally frozen ground, Tibetan Plateau is recognized as one of the most sensitive regions for climate change (Liu and Chen, 2000; Cheng and Wu, 2007; Yao et al., 2019). Monitoring and projecting the dynamics of hydrothermal and ecohydrological states and their responses to climate change on the Tibetan Plateau is important to help shed light on future ecosystem responses in this region. Considerable land-surface and vegetation changes have been reported in this region, e.g., degradation of permafrost and variations in seasonally frozen ground thickness (Cheng and Wu, 2007; Yao et al., 2019), advancing vegetation leaf onset dates (Zhang et al., 2013), and enhanced vegetation activity at the start of growing season (Qin et al., 2016). However, there are divergences with regard to the expected ecosystem changes across the Tibetan plateau (Cheng and Wu, 2007; Zhao et al., 2010; Qin et al., 2016; Wang et al., 2018). In response to climate warming, the degradation of frozen ground can positively affect the vegetation growth in mountainous region (Qin et al., 2016), but it can also lead to degradation of grasslands (Cheng and Wu, 2007), depending on soil hydrothermal regimes and climate conditions (Qin et al., 2016; Wang et al., 2016).

In this study, we investigated the consequences of considering coupled water and heat transfer processes on land-surface fluxes and ecosystem dynamics in the extreme environmental conditions of the Tibetan plateau, relying on land-surface and ecohydrological models confronted with multiple field observations. The inclusion or exclusion of different soil physical processes, i.e., explicitly considering the effect of soil ice content on hydrothermal properties and the tightly coupled water and heat transfer, in such environment frames the scope here. Specifically, the leading questions of the research are: i) How do different representations of frozen soil and coupled water and heat physics affect the simulated ecohydrological dynamics of a Tibetan plateau meadow? ii) How does different vadose zone physics affect our interpretation of mass, energy, and carbon fluxes in the ecosystem? Answering these two questions enables evaluation of the adequacy of models in simulating feedbacks among processes and ecosystem changes across the Tibetan plateau.

In order to achieve the aforementioned goals, the detailed soil mass and energy transfer scheme developed in the STEMMUS model (Zeng et al., 2011a, b; Zeng and Su, 2013) was incorporated into the ecohydrology model Tethys-Chloris (T&C) (Fatichi et al., 2012a, b). The frozen soil physics was explicitly taken into account and soil water and heat transfer were fully coupled to further facilitate the model's capability in dealing with complex vadose zone processes.

## 2. Experimental Site and Data

### 2.1 Experimental Site

In this study, we make use of the Maqu soil moisture and soil temperature (SMST) monitoring network (Su

et al., 2011; Dente et al., 2012; Su et al., 2013; Zeng et al., 2016), which is situated on the north-eastern fringe of the Tibetan Plateau. The monitoring network covers an area of approximately 40 km×80 km (33°30'–34°15'N, 101°38'–102°45'E) with the elevation varying from 3200 m to 4200 m above the sea level. The climate can be characterized by wet rainy summers and cold dry winters. The mean annual air temperature is 1.2 ℃ with about -10.0 ℃ and 11.7 ℃ for the coldest month (January) and warmest month (July), respectively. The alpine meadows (e.g., *Cyperaceae* and *Gramineae*) dominate in this region with a height of about 5 cm during the wintertime and 15 cm during the summertime. The general soil types are categorized as sandy loam, silt loam with a maximum of about 18% organic matter for the upper soil layers (Dente et al., 2012; Zheng et al., 2015a; Zheng et al., 2015b; Zhao et al., 2018). The groundwater level of the grassland area fluctuates from about 8.5 m to 12.0 m below the ground surface.

For the Maqu SMST monitoring network, SMST profiles are automatically measured by 5TM ECH$_2$O probes (METER Group, Inc., USA) at a 15-min interval. The meteorological forcing (including wind speed/direction, air temperature and relative humidity at five heights above ground) is recorded by a 20 m Planetary Boundary Layer (PBL) tower system. An eddy-covariance system (EC150, Campbell Scientific, Inc., USA) was installed for monitoring the dynamics of the turbulent heat fluxes and carbon fluxes. Instrumentations for measuring four-component down and upwelling solar and thermal radiation (NR01-L, Campbell Scientific, Inc., USA), and liquid precipitation (T200B, Geonor, Inc., USA) are also deployed.

For this research, data from March 2016 to August 2018 collected at the central experimental site (33°54'59"N, 102°09'32", elevation: 3430m) were utilized (see Figure 1). Seasonally frozen ground is characteristic of this site, with the maximum freezing depth approaching around 0.8 m under current climate conditions. The dedicated SMST profile (central station, Figure 1), with sensors installed at depths of 2.5 cm, 5 cm, 10 cm, 20 cm, 40 cm, 60 cm, and 100 cm, was used for validating the model simulations. Note that there are data gaps (25[th] Mar – 8[th] June, 2016; 29[th] Mar – 27[th] July, 2017, extended to 12[th] Aug, 2018 for 40 cm) due to the malfunction of instruments and the difficulty to maintain the network under such harsh environmental conditions.

**2.2 Data**

**2.2.1 Land Surface Energy and Carbon Fluxes and Vegetation Dynamics**

Starting from the raw *NEE* (Net Ecosystem Exchange) and ancillary meteorological data (friction velocity $u_*$, global radiation $R_g$, soil temperature $T_{soil}$, air temperature $T_{air}$, and vapor pressure deficit *VPD*), we employed the REddyProc package (Reichstein et al., 2005; Wutzler et al., 2018) as a post-processing tool to obtain the time series of *NEE*, *GPP* and ecosystem respiration $R_{eco}$ dynamics. Three different techniques, $u_*$ filtering, gap filling, and flux partitioning, were adopted in REddyProc package. The periods with low turbulent mixing was firstly determined and filtered for quality control ($u_*$ filtering, Papale et al., 2006). Then, considering the covariation of fluxes with meteorological variables and the temporal auto-correlation of fluxes, the marginal distribution sampling algorithm was used as the gap-filling method to replace the missing

data (Reichstein et al., 2005). Three cases were identified according to the availability of $R_g$, $T_{air}$, and $VPD$: Case 1, $R_g$, $T_{air}$, and $VPD$ data are available; Case 2, only $R_g$ data are available; Case 3, none of the $R_g$, $T_{air}$, and $VPD$ data are available. A look-up table (LUT) method was used to search for the similar meteorological conditions (i.e., under which $R_g$, $T_{air}$, and $VPD$ do not deviate by more than 50 W m$^{-2}$, 2.5 °C, and 5 hPa, respectively, for case 1) within a certain time window. The average value of *NEE* under these similar meteorological conditions was used to replace the missing gaps. The time window size started from 7 days and extended to 14 days if no similar meteorological conditions were detected. The similar LUT approach was utilized for case 2, the similar meteorological conditions were determined only by $R_g$ within a time window of 7 days. For case 3, the missing value of *NEE* was replaced by the average value of adjacent hours (within 1 hour) at the same day or at the same time of the day, which was derived from the mean diurnal course within 2 days. The aforementioned three steps were repeated with increased window sizes until the missing value could be properly filled. Finally, *NEE* was separated into *GPP* and $R_{eco}$ by night-time based and day-time based approaches (Lasslop et al., 2010). Land surface energy fluxes (*LE*, *H*) were processed simultaneously using the aforementioned $u_*$ filtering and gap filling methods with REddyProc package.

Furthermore, we downloaded MCD15A3H (Myneni et al., 2015) and MOD17A2H (Running et al., 2015) products for this site as the auxiliary ecosystem carbon and vegetation dynamics data, from the Oak Ridge National Laboratory Distributed Active Center (ORNL DAAC) website. MCD15A3H provides an estimation of 8-day composites of LAI (Leaf Area Index) and FAPAR (Fraction of Absorbed Photosynthetically Active Radiation), while MOD17A2H an 8-day composite of *GPP* (Gross Primary Production). Both MODIS products are at a resolution of 500m.

### 2.2.2 Precipitation, Evapotranspiration, and Frost Front

The observed surface water conditions over the entire study period, including the precipitation and cumulative evapotranspiration (which is obtained by summing up the hourly latent heat flux measured by EC system), are shown in Figure 2a. Both ET and precipitation are low until the end of the freezing period (see Figure 2b), during this early period the daily average ET is 0.15 mm/d. During the growing season, the cumulative precipitation increases and ET follows at a lower rate. The average daily ET for the entire observation period is 1.45 mm/d.

Figure 2b presents the development of freezing depth with time. Several freezing/thawing cycles frequently occurred at the beginning of the winter, which initializes the Freezing-Thawing (FT) process. The freezing front started to propagate with an average rate of 1.34 cm/d and 0.86 cm/d, reaching its maximum depth at around 80 cm and 70 cm for the year 2016-2017 and 2017-2018, respectively. Then the thawing process was activated by the atmospheric forcing at the surface and subsurface soil heat flux at the bottom of the soil.

## 3. Modelling the Soil-Plant-Atmosphere Continuum

### 3.1 T&C Model (unCPLD)

The Tethys-Chloris model (T&C) (Fatichi et al., 2012b) simulates the dynamics of energy, water, and vegetation and has been successfully applied to a very large spectrum of ecosystems and environmental conditions (Fatichi and Ivanov, 2014; Fatichi et al., 2016b; Pappas et al., 2016; Fatichi and Pappas, 2017; Mastrotheodoros et al., 2017). The model simulates the energy, water, and carbon exchanges between the land surface and the atmospheric surface layer accounting for aerodynamic, undercanopy, and leaf boundary layer resistances, as well as for stomatal and soil resistance. The model further describes vegetation physiological processes including photosynthesis, phenology, carbon allocation, and tissue turnover. Dynamics of water content in the soil profile in the plot-scale version are solved using the one-dimensional (1-D) Richards equation. Heat transfer in the soil is solved by means of the heat diffusion equation. Soil heat and water dynamics are uncoupled (however, note that T&C is termed unCPLD to distinguish it later with the coupling with STEMMUS). The detailed model description is provided in the above-mentioned references and some key elements applied for this study are explained in the following.

T&C model uses the 1-D Richards equation, which describes the water flow under gravity and capillary forces in isothermal conditions for variably saturated soils:

$$\rho_L \frac{\partial \theta}{\partial t} = -\frac{\partial q}{\partial z} - S = \rho_L \frac{\partial}{\partial z}\left[K\left(\frac{\partial \psi}{\partial z} + 1\right)\right] - S \tag{1}$$

where $\theta$ (m$^3$ m$^{-3}$) is the volumetric water content; $q$ (kg m$^{-2}$ s$^{-1}$) is the water flux; $z$ (m) is the vertical direction coordinate; $S$ (kg m$^{-3}$ s$^{-1}$) is the sink term for transpiration and evaporation fluxes. $\rho_L$ (kg m$^{-3}$) is the liquid water density; $K$ (m s$^{-1}$) is the soil hydraulic conductivity; $\psi$ (m) is the soil water potential; $t$ (s) is the time. In T&C, the nonlinear partial differential equation is solved using a finite volume approach with the method of lines (MOL) (Lee et al., 2004). MOL discretizes the spatial domain and reduces the partial differential equation to a system of ordinary differential equations in time, which can be expressed as:

$$d_{z,i}\frac{d\theta_i}{dt} = q_{i-1} - q_i - T_v r_{v,i} - E_s - E_{bare} \tag{2}$$

where $d_{z,i}$ (m) is the thickness of layer $i$; $q_i$ (m s$^{-1}$) is the vertical outflow from a layer $i$; $T_v$ (m s$^{-1}$) is the transpiration fluxes from the vegetation; $r_{v,i}$ is the fraction of root biomass contained in soil layer $i$; $E_{bare}$ (m s$^{-1}$), evaporation from the bare soil; $E_s$ (m s$^{-1}$), evaporation from soil under the canopy.

The heat conservation equation used in the T&C neglects the coupling of water and heat transfer physics and only the heat conduction component is considered, which can be expressed as:

$$\rho_{soil} C_{soil} \frac{\partial T}{\partial t} = \frac{\partial}{\partial z}\left(\lambda_{eff} \frac{\partial T}{\partial z}\right) \tag{3}$$

where $\rho_{soil}$ (kg m$^{-3}$) is the bulk soil density; $C_{soil}$ (J kg$^{-1}$ K$^{-1}$) is the specific heat capacities of bulk soil; $\lambda_{eff}$ (W m$^{-1}$ K$^{-1}$) is the effective thermal conductivity of the soil. $T$ (K) is the soil temperature. When soil undergoes freezing/thawing process, the latent heat flux due to water phase change becomes important, which

is not considered in the original T&C model, but it is in the T&C-FT (freezing/thawing) model.

### 3.2 T&C-FT Model (unCPLD-FT)

To account for frozen soil physics, T&C-FT model considers ice effect on hydraulic conductivity, thermal conductivity, heat capacity, and subsurface latent heat flux. However, the vapor flow and the thermal effect on water viscosity are not considered in T&C-FT, and during the non-frozen period, soil water and heat are
still independently transferred as in T&C (this version is named here unCPLD-FT). To explicitly account for freezing/thawing processes, the heat conservation equation is written as

$$\rho_{soil}C_{soil}\frac{\partial T}{\partial t} - \rho_{ice}L_f\frac{\partial \theta_{ice}}{\partial t} = \frac{\partial}{\partial z}\left(\lambda_{eff}\frac{\partial T}{\partial z}\right) \tag{4}$$

where the latent heat associated with the freezing/thawing process is explicitly considered and ice water content $\theta_{ice}$ is a prognostic variable, which is simulated along with liquid water content for each soil layer. Specifically, when Eq. (4) is rewritten in terms of an apparent volumetric heat capacity $C_{app}$ (Hansson et al.,
2004; Gouttevin et al., 2012), it can be solved equivalently to Eq. (3):

$$C_{app}\frac{\partial T}{\partial t} = \frac{\partial}{\partial z}\left(\lambda_{eff}\frac{\partial T}{\partial z}\right) \tag{5}$$

where $C_{app}$ can be computed knowing the temperature $T$ (K), latent heat of fusion $L_f$ and the differential (specific) water capacity $d\theta/d\psi$ at a given liquid water content $\theta$ (Hansson et al., 2004):

$$C_{app} = \rho_{soil}C_{soil} + \rho_{ice}\frac{L_f^2}{gT}\frac{d\theta}{d\psi} \tag{6}$$

The effective thermal conductivity $\lambda_{eff}$ (W m$^{-1}$ K$^{-1}$) and the specific soil heat capacity $C_{soil}$ (J kg$^{-1}$ K$^{-1}$) are computed accounting for solid particles, water, and ice content (Johansen, 1975; Farouki, 1981; Lawrence et
al., 2018; Yu et al., 2018). The soil freezing characteristic curve providing the liquid water potential in a frozen soil is computed following the energy conservative solution proposed by Dall'Amico et al. (2011) and it can be combined with various soil hydraulic parameterizations including van Genuchten and Saxton and Rawls to compute the maximum liquid water content at a given temperature and consequently ice and liquid content profiles at any time step (Fuchs et al., 1978; Yu et al., 2018).

Finally, saturated hydraulic conductivity is corrected in the presence of ice content (e.g., Hansson et al., 2004; Yu et al., 2018). Note that beyond latent heat associated with phase change and changes in thermal and hydraulic parameters because of ice presence, all the other soil physics processes described by STEMMUS are not considered here, and heat and water fluxes are still not entirely coupled in T&C-FT.

### 3.3 STEMMUS Model

**S**imultaneous **T**ransfer of **E**nergy, **M**ass and **M**omentum in **U**nsaturated **S**oil (STEMMUS) model solves soil water and soil heat balance equations simultaneously in one time step (Zeng et al., 2011a, b; Zeng and Su, 2013). The Richards' equation with modifications made by Milly (1982) is utilized to mimic the coupled soil mass and energy transfer process. The vapor diffusion, advection, and dispersion are all taken into account

as water vapor transport mechanisms. The root water uptake process is regarded as the sink term of soil water and heat balance equations, building up the linkage between soil and atmosphere (Yu et al., 2016). In STEMMUS, temporal dynamics of three phases of water (liquid, vapor and ice) are explicitly presented and simultaneously solved by spatially discretizing the corresponding governing equations of liquid water flow and vapor flow.

$$\frac{\partial}{\partial t}(\rho_L \theta_L + \rho_V \theta_V + \rho_{ice}\theta_{ice}) = -\frac{\partial}{\partial z}(q_{Lh} + q_{LT} + q_{Vh} + q_{VT}) - S$$
$$= \rho_L \frac{\partial}{\partial z}\left[K\left(\frac{\partial \psi}{\partial z} + 1\right) + D_{TD}\frac{\partial T}{\partial z}\right] + \frac{\partial}{\partial z}\left[D_{Vh}\frac{\partial \psi}{\partial z} + D_{VT}\frac{\partial T}{\partial z}\right] - S \tag{7}$$

where $\rho_V$ and $\rho_{ice}$ (kg m$^{-3}$) are the density of water vapor and ice, respectively; $\theta_L$, $\theta_V$ and $\theta_{ice}$ (m$^3$ m$^{-3}$) are the soil liquid, vapor and ice volumetric water content, respectively; $q_{Lh}$ and $q_{LT}$ (kg m$^{-2}$ s$^{-1}$) are the soil liquid water flow driven by the gradient of soil matric potential $\frac{\partial \psi}{\partial z}$ and temperature $\frac{\partial T}{\partial z}$, respectively. $q_{Vh}$ and $q_{VT}$ (kg m$^{-2}$ s$^{-1}$) are the soil water vapor fluxes driven by the gradient of soil matric potential $\frac{\partial \psi}{\partial z}$ and temperature $\frac{\partial T}{\partial z}$, respectively. $D_{TD}$ (kg m$^{-1}$ s$^{-1}$ K$^{-1}$) is the transport coefficient of the adsorbed liquid flow due to temperature gradient; $D_{Vh}$ (kg m$^{-2}$ s$^{-1}$) is the isothermal vapor conductivity; and $D_{VT}$ (kg m$^{-1}$ s$^{-1}$ K$^{-1}$) is the thermal vapor diffusion coefficient.

STEMMUS takes into account different heat transfer mechanisms, including heat conduction ($\lambda_{eff}\frac{\partial T}{\partial z}$), convective heat transferred by liquid and vapor flow, the latent heat of vaporization ($\rho_V \theta_V L_0$), the latent heat of freezing/thawing ($-\rho_{ice}\theta_{ice}L_f$) and a source term associated with the exothermic process of wetting of a porous medium (integral heat of wetting) ($-\rho_L W \frac{\partial \theta_L}{\partial t}$).

$$\frac{\partial}{\partial t}\left[(\rho_s \theta_s C_s + \rho_L \theta_L C_L + \rho_V \theta_V C_V + \rho_{ice}\theta_{ice}C_{ice})(T - T_{ref}) + \rho_V \theta_V L_0 - \rho_{ice}\theta_{ice}L_f\right] - \rho_L W \frac{\partial \theta_L}{\partial t}$$
$$= \frac{\partial}{\partial z}\left(\lambda_{eff}\frac{\partial T}{\partial z}\right) - \frac{\partial}{\partial z}[q_L C_L(T - T_{ref}) + q_V(L_0 + C_V(T - T_{ref}))] - C_L S(T - T_{ref}) \tag{8}$$

where $\rho_s$ (kg m$^{-3}$) is the soil solids density; $\theta_s$ is the volumetric fraction of solids in the soil; $C_s$, $C_L$, $C_V$ and $C_{ice}$ (J kg$^{-1}$ K$^{-1}$) are the specific heat capacities of soil solids, liquid, water vapor and ice, respectively; $T_{ref}$ (K) is the arbitrary reference temperature; $L_0$ (J kg$^{-1}$) is the latent heat of vaporization of water at the reference temperature; $L_f$ (J kg$^{-1}$) is the latent heat of fusion; $W$ (J kg$^{-1}$) is the differential heat of wetting (expressed by Edlefsen and Anderson (1943) as the amount of heat released when a small amount of free water is added to the soil matrix). $q_L$ and $q_V$ (kg m$^{-2}$ s$^{-1}$) are the liquid and vapor water flux, respectively. Additional details on the equations for solving the coupled water and heat equations can be found in Zeng et al. (2011a, b) and Zeng and Su (2013). In the appendix, a notation table is summarized for the above equations.

**3.4 Coupling T&C and STEMMUS (CPLD)**

As mentioned above (section 3.1-3.2), T&C considers soil water and heat dynamics independently, and T&C-FT only considers ice effects associated with latent heat, thermal and hydraulic parameters, while all other

soil physics processes of STEMMUS are not considered. On the other hand, while STEMMUS model can well reproduce the soil water and heat transfer process in frozen soil, it lacks a detailed description of land-surface processes and of the ecohydrological feedback mechanisms. To take advantage of the strengths of both models, we coupled STEMMUS model with the land-surface and vegetation components of T&C model (termed as CPLD) to better describe the soil-plant-atmosphere continuum (SPAC) in cold regions.

The current coupling procedure between STEMMUS and the T&C model is based on a sequential coupling via exchanging mutual information within one time step (see Figure 3). T&C model and STEMMUS model run sequentially within one time step. First, the preparation and initialization modules are called. Meteorology inputs and constant parameters are set, and the initialization process is performed. After the inputs are prepared, the main iteration process starts. T&C is in charge of the time control information (starting time, time step, elapsed time) and informs STEMMUS model with these time settings every time step. Meanwhile, the surface boundary conditions obtained by the solution of vegetation and land-surface energy dynamics are also sent to drive STEMMUS model. The surface latent heat flux (*LE*) is partitioned into soil evaporation (used for setting the surface boundary condition of soil water flow) and plant transpiration (further subdivided into layer-specific root water uptakes representing the sink terms of Richard equation).

After convergence is achieved in the soil module (i.e., convergence criteria is set as 0.001 for both soil matric potential [in cm] and soil temperature [in K]), STEMMUS estimates soil temperature/soil moisture (hereafter as ST/SM) profiles, which are utilized to update ST/SM states in T&C model. T&C model then utilizes these updated ST/SM information (rather than its own computed ST/SM profiles) to proceed with the ecohydrological simulations in the following time step. Such iteration continues till the end of simulation period.

### 3.5 Numerical Experiments

To investigate the role of increasing complexity of vadose zone physics in ecosystem functioning, three numerical experiments were designed on the basis of the aforementioned modeling framework (Table 1). First experiment, the T&C original model was run as stand-alone, termed as unCPLD simulation. For the unCPLD model, soil water and heat transfer is independent with no explicit consideration of soil ice effect. The second experiment, the updated T&C model with explicit consideration of freezing/thawing process was run as it can estimate the dynamics of soil ice content and the related effect on water and heat transfer (e.g., blocking effect on water flow, heat release/gain due to phase change) but otherwise being exactly equal to T&C original model. This second simulation is named the unCPLD-FT simulation, where the term unCPLD generally refers to the fact that T&C model and STEMMUS model are not yet coupled. The third experiment, STEMMUS model was coupled with T&C model to enable not only frozen soil physics but also additional processes and most importantly the tight coupling of water and heat effects. This simulation is named CPLD simulation. In this third scenario, vapor flow, which links the soil water and heat flow, is explicitly considered. In addition to the ice blocking effect as presented in unCPLD-FT, the thermal effect on water flow is also expressed with the temperature dependence of hydraulic conductivity and matric potential. Furthermore, not

only the latent heat due to phase change, but also the convective heat due to liquid/vapor flow is simulated.

Hourly meteorological forcing (including downwelling solar and thermal radiation, precipitation, air temperature, relative humidity, wind speed, atmospheric pressure) was utilized to drive the models. For the adaptive time step of STEMMUS simulation, the linear interpolation between two adjacent hourly meteorological measurements was used to generate the required values at every second. The hydrological related initial states, e.g., initial snow water equivalent, soil water and temperature profiles, were taken as close as possible to the observed ones. Since the current initial conditions of the carbon and nutrients pools in the soil are unknown, we spin-up carbon and nutrient pools running only the soil-biogeochemistry module for 1000 years using average climatic conditions and prescribed litter inputs taken from preliminary simulations. Then we used the spun-up pools as initial conditions for the hourly-scale simulation over the period for which hourly observations are available. This last operation is repeated two times, which allows reaching a dynamic equilibrium of nutrient and carbon pools in the soil and vegetation.

The total depth of soil column was set as 3m and divided into 18 layers with a finer discretization in the upper soil layers (1-5cm) than that in the lower soil layers (10-50cm). Soil samples were collected and transported to the laboratory to determine the soil hydrothermal properties (see Zhao et al. 2018 for detail). The average soil texture and fitted Van Genuchten parameters at three soil layers were listed in supplement Table S1. Vegetation parameters were obtained on the basis of literature and expert knowledge (see a summary of the adopted vegetation parameters in the supplement Table S2). All three numerical experiments shared the same soil and vegetation parameter settings.

## 4. Results and Discussion

### 4.1 Surface Fluxes Simulations

The 5-day moving average dynamics of the net incoming radiation ($Rn$), latent heat ($LE$) and sensible heat ($H$) fluxes measured and simulated by the unCPLD model, unCPLD-FT, and CPLD model for the study period are presented in Figure 4. The seasonality and magnitude of surface fluxes can be captured across seasons. A good match between observed and simulated $Rn$ and $LE$ was identified during the whole period, with isolated observable discrepancies (Figure 4a & 4c and Figure S1). Compared with unCPLD and unCPLD-FT simulations, CPLD model simulated similar dynamics of $LE$ while it generally produced a larger overestimation of $Rn$, especially during the frozen period. These mismatches of $Rn$ can be partly attributed to the uncertainties of observed winter precipitation events and the following snow cover dynamics, which might not be well captured in the models. For the sensible heat flux simulations, all three models can reproduce the seasonal dynamics. However, an overestimation of the 5-day average values was observed in several periods. Given the good correspondence between observations and simulations of net radiation and latent heat, this discrepancy might be a model shortcoming due to the simplification in considering only one single surface prognostic temperature (i.e. soil surface and vegetation surface temperature were assumed the same), but it can be also caused by the lack of energy balance closure in the eddy-covariance data (see Sect.

4.5). Compared with unCPLD and unCPLD-FT simulations, the overestimation was reduced in the CPLD model simulations and the *H* dynamics was closer to observations during the growing season.

The correlation between observed and simulated daily average surface heat fluxes with unCPLD, unCPLD-FT, and CPLD model is shown in Figure 5 and Figure S2 and S3. Noticeably all the unCPLD/CPLD model scenarios, with different water and heat transfer physics, exhibited nearly identical statistical performance of surface fluxes simulations (Figure 5). The overall performance of the model in terms of turbulent flux simulations can be regarded as acceptable given the uncertainties in winter precipitation and eddy-covariance observations in such a challenging environment, even though discrepancies exist during certain periods (Figure 4).

**4.2 Soil Moisture and Soil Temperature Simulations**

The capability of the three models to reproduce the temporal dynamics of soil moisture is illustrated in Figure 6. By explicitly considering soil ice content, the unCPLD-FT and CPLD model captured well the response of soil moisture dynamics to the freeze-thaw cycles, while the unCPLD model lacked such capability and maintained a higher soil water content throughout the winter period, but slightly lower water content in the growing season. For all three models, the consistency between the measured and simulated soil water content at five soil layers was satisfactory during the growing season, indicating the models' capability in portraying the effect of precipitation and root water uptake on the soil moisture conditions.

Five layers of soil temperature measurements were employed to test the performance of the model in reproducing the soil temperature profiles (Figure 7). During the growing period, all three models can capture well the dynamics of soil temperature. In this period, there is no significant difference among the three models about the magnitude and temporal dynamics of soil temperature. During the freezing period, a general underestimation of soil temperature and overestimation of its diurnal fluctuations were found at shallower soil layers, which may indicate that there is some thermal buffering effect in reality not fully captured in the models. Compared with unCPLD-FT and CPLD models, the unCPLD model simulations had stronger diurnal fluctuations of soil temperature with an underestimation of temperature at the beginning of the freezing period and a considerable overestimation during the thawing phase. This results in an earlier date passing the 0°C threshold than in the unCPLD-FT and CPLD simulations. It should be noted that for the deeper soil layers (e.g., 60cm in Figure 7), all models tended to simulate the early start of freezing soil temperatures and considerably underestimated the soil temperature during the frozen period. This can be due to the uncertainties in soil organic layer parameters, the not fully captured snow cover effect (Gouttevin et al., 2012), a potentially pronounced heterogeneity in soil hydrothermal properties, or the potential role of solutes on the freezing-point depression (as the presence of solute lowers the freezing soil temperature) (Painter and Karra, 2014). These mismatches in deep soil temperature degraded the model performance in simulating the dynamics of liquid water (Figure 6) and ice content (Figure 8) during the frozen period.

**4.3 Soil Ice Content and Water Flux**

The time-series of soil ice content and water flux from unCPLD, unCPLD-FT and CPLD model simulations for soil layers below 2 cm are presented in Figure 8. As soil ice content measurements were not available, the freezing front propagation inferred from the soil temperature measurements was employed to qualitatively

assess the model performance. The phenomenon that a certain amount of liquid water flux moves upwards along with the freezing front can be clearly noticed for both the unCPLD-FT and CPLD model simulations. As the soil matric potential changes sharply during the water phase change, a certain amount of water fluxes will be forced towards the phase changing region, a phenomenon known as cryosuction. Such a phenomenon has already been demonstrated from theoretical and experimental perspectives by many researchers (Hansson

et al., 2004; Watanabe et al., 2011; Yu et al., 2018; Yu et al., 2020). Cryosuction is much more accentuated in the unCPLD-FT simulation, while it is of course absent in the unCPLD model simulations (Figure 8c). Precipitation induced downward water flux can be observed in all models during summer with very similar patterns. It is to note that compared to unCPLD-FT model, CPLD model presented a relatively lower presence of soil ice content, while its temporal dynamics was closer to the observed freezing/thawing front propagation.

The difference between the two simulations can be attributed to the constraints imposed by the interdependence of liquid, ice, and vapor in the soil pores that is considered only in CPLD model.

**4.4 Simulations of Land Surface Carbon Fluxes**

The eddy covariance derived vegetation productivity and remote sensing (MODIS) observations of vegetation dynamics are compared with the model simulation in Figure 9. When compared with in situ eddy-

covariance observations, slightly earlier growth and considerably earlier senescence of grassland with lower photosynthesis were inferred from MODIS *GPP* product (Figure 9a). The mismatch in the phenology is likely a combined issue of 8-day (or longer if clouds are impeding the view) composite of MODIS products and challenge of translating vegetation reflectance signals into productivity or Leaf Area Index (LAI) during the grass senescent phase.

Taking eddy-covariance observations as the reference, the onset date of grassland appears to be well captured by both unCPLD and CPLD model simulations, while a delayed onset date by unCPLD-FT model. Leaf senescence and dormancy phase are a bit delayed in the models when compared with eddy-covariance data and considerably delayed when compared to MODIS-LAI, even though the latter is particularly uncertain as described above. Although there is an observable underestimation of *GPP* compared to the eddy covariance

measurements, the dynamics of *GPP*, which is mainly constrained by the photosynthetic activity and environmental stresses, is reasonably reproduced by all model simulations.

The underestimation of *GPP* has magnified consequences in terms of reproducing *NEE* dynamics by unCPLD and CPLD models. While this might be seen as a model shortcoming, there are a number of reasons that lead to questioning the reliability of the magnitude of carbon fluxes measurements at this site. By

checking other ecosystem productivity under similar conditions, the annual average *GPP* for the Tibetan

plateau meadow ecosystem ranges from 300 to 935 g C m$^{-2}$ yr$^{-1}$, while the annual average *NEE* ranges from -79 to -213 g C m$^{-2}$ yr$^{-1}$ (see the literature summary in the Supplement Table S3). While the EC system used in this experimental site observes an annual *GPP* and *NEE* as 1132.52 and -293.24 g C m$^{-2}$ yr$^{-1}$. Both the *GPP* and *NEE* measured fluxes are significantly larger than existing estimates of the carbon exchange for

such ecosystem type and are unlikely to be correct in absolute magnitude. The ecosystem respiration ($R_{eco}$), indicating the respiration of activity of all living organisms in an ecosystem is shown in Figure 9d. The performance of all three model simulations in reproducing $R_{eco}$ dynamics can be characterized as an overall good match with regards to the magnitude and seasonal dynamics, which further suggests the discrepancy in observed/simulated *GPP* is the driver of the disagreement in *NEE.*

The difference in the soil liquid water and temperature profile simulations between the CPLD and unCPLD models (as shown in Figures 6 & 7) resulted in differences in simulated vegetation dynamics, especially concerning the leaf onset date, which is affected by integrated winter soil temperatures. The unCPLD-FT model has a delay in the vegetation onset date when compared to other simulations, due to the significant cryosuction that prolongs freezing conditions and keeps lower soil temperatures. This makes the unCPLD

simulation having slightly shorter vegetation active season compared to the CPLD model simulations. The lower *GPP* in the unCPLD simulations is instead related to a slightly enhanced water-stress induced by the different soil-moisture dynamics during the winter and summer season with a lower root zone moisture produced by the unCPLD model (Figure 6), which affects the plant photosynthesis and growth. Differences in soil temperature profiles can also affect root respiration in generating additional small differences in *GPP*.

**4.5 Surface Energy Balance Closure**

The energy balance closure problem, usually identified because the sum of latent (*LE*) and sensible (*H*) heat fluxes is less than the available energy (*Rn-G$_0$*), is quite common in eddy covariance measurements (Su, 2002; Wilson et al., 2002; Leuning et al., 2012). The energy imbalance of EC measurements is particularly significant at sites over the Tibetan Plateau (Tanaka et al., 2003; Yang et al., 2004; Chen et al., 2013; Zheng

et al., 2014). Figure 10 presents the energy imbalance of hourly *LE* and *H* by the eddy covariance measurements, observed *Rn* by the four-component radiation measurements, and the estimated ground heat flux (*G$_0$*) by CPLD model. The sum of measured *LE* and *H* was significantly less than *Rn*, with the slope of *LE+H* versus *Rn* equal to 0.59 (Figure 10a). Usually, the measurements of radiation are reliable (Yang et al., 2004). If we assume that the turbulence fluxes (*LE*, *H*) measurements are accurate, then the rest of energy

(around 41% of *Rn*) should be theoretically consumed by ground heat flux *G$_0$*, which is clearly impossible. When compared to the available energy (*Rn-G$_0$*), the slope was increased to 0.70 (Figure 10b). Table 2 demonstrated that the energy imbalance problem was significant across all seasons. The seasonal variation of energy closure ratio (ECR) can be identified for the case *LE+H* versus *Rn-G$_0$*, similar to the research of Tanaka et al. (2003), i.e., a good energy closure during the pre-monsoon periods while a degraded one during

the summer monsoon periods.

These problems clearly suggest that care should be taken to the data mutual corroboration issue. Nevertheless, such issue is not affecting the comparison results among models with different vadose zone physics, since we did not force any parameter calibration or data-fitting procedure, but simply rely on physical constraints, literature, and expert knowledge to assign model parameters.

**4.6 Effects on Water Budget Components**

The effect of different model versions on soil water budget components is illustrated in Figure 11. T&C model can describe in detail different water budget components. Precipitation can be partitioned into vegetation interception, surface runoff, and infiltration. Infiltrated water can then be used for surface evaporation ($Es$), root water uptake (i.e., transpiration, $Tv$), and changes in soil water storage ($\Delta Vs$). The other evaporation components, i.e., evaporation from intercepted canopy water ($E_{IN}$) and snow cover ($E_{SN}$), can be further distinguished by T&C model. A certain amount of water will drain below the bottom of the 3 m soil column as deep leakage ($L_K$).

All models demonstrated that most of the precipitation is used by ET. Less amount of water was consumed by ET from unCPLD-FT simulations than that from unCPLD. This is due to the lower amount of vegetation transpiration ($Tv$) and intercepted canopy water evaporation ($E_{IN}$) regulated by cooler late winter temperatures and the late beginning of the active vegetation season. The cooler late winter temperatures from unCPLD-FT simulations can be attributed to the retardation of the thawing process due to the phase change-induced heat absorption and the soil ice-induced modification of bulk heat capacity during the freezing-thawing transition period, which damped the magnitude of temperature variations and delayed the thawing process. With explicit consideration of soil ice, hydraulic conductivity is also reduced and vertical water flow is retarded during the frozen period (Kurylyk and Watanabe, 2013). This explains the higher value of $\Delta Vs$ of unCPLD-FT simulation (5.2%) than that of unCPLD simulation (2.8%). Furthermore, at the end of the freezing period, the unCPLD-FT simulation presents a delayed vegetation onset thus a decrease of ecosystem water consumption, which favors percolation toward deeper layers and the bottom leakage. Such a positive effect on the bottom leakage flux was slightly weaker than the negative effect (impeded water flow) due to frozen soil throughout the winter season. These results indicate that the presence of seasonally frozen soil can mediate the water storage in the vadose zone via both hydrological and plant physiological controls.

The effect of coupled water and heat physics (unCPLD vs. CPLD model) on the water budget components can be summarized as: i) the amount of ecosystem water consumption ET was reduced, due to the damped surface evaporation process (evaporation from the soil surface and intercepted water). ii) water storage amount in the vadose zone increased while the bottom leakage decreased. We attribute this to the way ice content is simulated in the CPLD simulation, and also to the temperature dependence of soil hydraulic conductivity (see Table 1 and Supplement S1). Specifically, the high accumulation of ice content in the unCPLD-FT simulations indicates a relatively stronger cryosuction effect than in CPLD simulations. This cryosuction effect is mitigated in the fully coupled model because of water vapor transfer and thermal gradients, even though different solutions in the parameterization of bulk soil thermal conductivity and

volumetric soil heat capacity could also be responsible for the difference (Yu et al., 2018; Yu et al., 2020). Overall, taking into account the fully coupled water and heat physics modifies the temporal dynamics of ice formation and thawing in the soil and activates temperature effects on water flow (i.e., low soil temperature will slow down water movement).

### 4.7. The Influence of Different Mass/Heat Transfer Processes

Given the same atmospheric forcing and the same model structure to represent land-surface exchanges and vegetation dynamics, different vadose zone physics generates differences in SM and ST vertical profiles. From the perspective of energy fluxes, the convective heat flux and explicit frozen soil physics are taken into account in the CPLD model, while they are not considered in the two unCPLD models. The difference among models in simulating the liquid water flux-induced convective heat flux is mostly relevant to the freezing or thawing process (Kane et al., 2001; Boike et al., 2008; Sjöberg et al., 2016; Chen et al., 2020; Yu et al., 2020). As it has been observed, a certain amount of liquid water/vapor flux moves toward the freezing front and this effect is different between unCPLD-FT and CPLD while absent in unCPLD (Figure 8). For the unfrozen period, instead, the total mass fluxes were comparable between the two unCPLD and CPLD simulations. For the temperature gradient, there is not much difference between unCPLD and CPLD simulations during both the growing season and freezing-thawing period. The latent heat released by freezing and consumed by the melting processes slows down the freezing/thawing process and decreases the diurnal and seasonal temperature fluctuations (Figure 7). Different soil thermal profiles have consequences on the vegetation dynamic process (Figure 9), mainly by affecting the beginning of the growing season and the subsequent simulated photosynthesis and growth processes. This is consistent with the decadal observation results of Li et al. (2016), in which they reported the cumulative temperature effect on the carbon dynamics as it breaks the vegetation dormancy, affects the leaf phenology and plant growth dynamics. From the perspective of water fluxes, it is during the frozen period that water and heat transfer processes are tightly coupled (Hansson et al., 2004; Yu et al., 2018; Yu et al., 2020). Both the explicit consideration of soil ice and coupled water and heat physics can affect the vadose zone water flow via altering the hydraulic conductivity and soil water potential gradients. This is testified by the fact that the unCPLD-FT simulation accounting for soil-freezing in a simplified way, in comparison to STEMMUS (e.g., the CPLD simulation), cannot recover the exact dynamics of ice content (Figure 8), which impacts leaf onset and to a less extent hydrological fluxes. However, in the rest of the year, the simplified solution of vadose zone physics of T&C leads to very similar results as the coupled one, suggesting that most of the additional physics does not modify substantially the ecohydrological response during unfrozen periods.

### 5 Conclusion

The detailed vadose zone process model STEMMUS and the ecohydrological model T&C were coupled to investigate the effect of various model representations in simulating water and energy transfer and seasonal ecohydrological dynamics over a typical Tibetan meadow. The results indicate that the original T&C model

tended to overestimate the variability and magnitude of soil temperature during the freezing period and the freezing-thawing transition period. Such mismatches were ameliorated by the inclusion of soil ice content and freezing-thawing processes to the original model, and further improved with explicit consideration of coupled water and heat physics. For the largest part of the simulated period (i.e., unfrozen), we found that a simplified treatment of vadose zone dynamics is sufficient to reproduce satisfactory energy, water and carbon fluxes – given the uncertainty in the eddy-covariance observations. Additional complexity in vadose zone representation is mostly significant during the freezing and thawing periods as ice content simulations differ among models and the amount of water moving towards the freezing front was differently simulated. These discrepancies have an impact (even though limited to the beginning of the growing season) on vegetation dynamics. The leaf onset is better captured by the unCPLD and CPLD models, while a delayed onset date was reproduced by unCPLD-FT model. Nonetheless, overall patterns for the rest of the year do not differ considerably among simulations, which suggests that the difference in vadose zone dynamics, by using a fully coupled water-heat model treatment, is not enough to affect the overall ecosystem response. This also suggests that the additional complexity might be more needed for specific vadose zone studies and investigation of permafrost thawing rather than for ecohydrological applications. Nevertheless, the coupled model can reveal the hidden physically-based processes and mechanisms in the vadose zone that cannot be explained by uncoupled models, which can assist the comprehensive physical interpretations of ecosystem responses to subtle climatic changes/trends over high-altitude cold regions. In summary, our investigations using different models of vadose zone physics can be helpful to support the development and application of earth system models as they suggest that a certain degree of complexity might be necessary for specific analyses.

*Data availability.* The soil hydraulic/thermal property data can be accessed from 4TU. Center for Research Data (https://doi.org/10.4121/uuid:61db65b1-b2aa-4ada-b41e-61ef70e57e4a). The other relevant data are available from https://doi.org/10.6084/m9.figshare.12058038.v1 or from Data Archiving and Networked Services (DANS) https://easy.dans.knaw.nl/ui/datasets/id/easy-dataset:160877 upon registration.

*Author contribution.* ZS and YZ conceptualized the study, LY, YZ, and SF developed the methodology and prepared the original draft of the paper, LY, YZ, SF, and ZS all contributed to the reviewing and editing of the final paper.

*Competing interests.* The authors declare that they have no conflict of interest.

**Acknowledgment**

This work is supported by the National Natural Science Foundation of China (grant no. 41971033) and supported by the Fundamental Research Funds for the Central Universities, CHD (grant no. 300102298307).

The authors would like to thank the editor and referees for their constructive comments and suggestions on improving the manuscript.

**Notation**

| Symbol | Parameter | Unit |
|--------|-----------|------|
| $C_{app}$ | Apparent heat capacity | $\mathrm{J\,kg^{-1}\,K^{-1}}$ |
| $C_{ice}$ | Specific heat capacity of ice | $\mathrm{J\,kg^{-1}\,K^{-1}}$ |
| $C_L$ | Specific heat capacity of liquid | $\mathrm{J\,kg^{-1}\,K^{-1}}$ |
| $C_s$ | Specific heat capacity of soil solids | $\mathrm{J\,kg^{-1}\,K^{-1}}$ |
| $C_{soil}$ | Specific heat capacity of the bulk soil | $\mathrm{J\,kg^{-1}\,K^{-1}}$ |
| $C_V$ | Specific heat capacity of water vapor | $\mathrm{J\,kg^{-1}\,K^{-1}}$ |
| $d_{z,i}$ | Thickness of layer $i$ | $\mathrm{m}$ |
| $D_{Vh}$ | Isothermal vapor conductivity | $\mathrm{kg\,m^{-2}\,s^{-1}}$ |
| $D_{VT}$ | Thermal vapor diffusion coefficient | $\mathrm{kg\,m^{-1}\,s^{-1}\,K^{-1}}$ |
| $D_{TD}$ | Transport coefficient for adsorbed liquid flow due to temperature gradient | $\mathrm{kg\,m^{-1}\,s^{-1}\,K^{-1}}$ |
| $E_{bare}$ | Evaporation from the bare soil | $\mathrm{m\,s^{-1}}$ |
| $E_s$ | Evaporation from soil under the canopy | $\mathrm{m\,s^{-1}}$ |
| $K$ | Hydraulic conductivity | $\mathrm{m\,s^{-1}}$ |
| $K_s$ | Soil saturated hydraulic conductivity | $\mathrm{m\,s^{-1}}$ |
| $L_f$ | Latent heat of fusion | $\mathrm{J\,kg^{-1}}$ |
| $L_0$ | Latent heat of vaporization of water at the reference temperature | $\mathrm{J\,kg^{-1}}$ |
| $n$ | Van Genuchten fitting parameters | - |
| $q$ | Water flux | $\mathrm{kg\,m^{-2}\,s^{-1}}$ |
| $q_i$ | Vertical outflow from a layer $i$ | $\mathrm{m\,s^{-1}}$ |
| $q_L$ | Soil liquid water fluxes (positive upwards) | $\mathrm{kg\,m^{-2}\,s^{-1}}$ |
| $q_V$ | Soil water vapor fluxes (positive upwards) | $\mathrm{kg\,m^{-2}\,s^{-1}}$ |
| $q_{Lh}$ | Liquid water flux driven by the gradient of matric potential | $\mathrm{kg\,m^{-2}\,s^{-1}}$ |
| $q_{LT}$ | Liquid water flux driven by the gradient of temperature | $\mathrm{kg\,m^{-2}\,s^{-1}}$ |
| $q_{Vh}$ | Water vapor flux driven by the gradient of matric potential | $\mathrm{kg\,m^{-2}\,s^{-1}}$ |
| $q_{VT}$ | Water vapor flux driven by the gradient of temperature | $\mathrm{kg\,m^{-2}\,s^{-1}}$ |
| $r_{v,i}$ | Fraction of root biomass contained in soil layer $i$ | - |
| $S$ | Sink term for transpiration, evaporation | $\mathrm{kg\,m^{-3}\,s^{-1}}$ |
| $t$ | Time | $\mathrm{s}$ |
| $T$ | Soil temperature | $\mathrm{K}$ |
| $T_{ref}$ | Arbitrary reference temperature | $\mathrm{K}$ |
| $T_v$ | Transpiration fluxes from the vegetation | $\mathrm{m\,s^{-1}}$ |
| $W$ | Differential heat of wetting | $\mathrm{J\,kg^{-1}}$ |
| $z$ | Vertical space coordinate (positive upwards) | $\mathrm{m}$ |
| $\alpha$ | Air entry value of soil | $\mathrm{m^{-1}}$ |
| $\psi$ | Water potential | $\mathrm{m}$ |
| $\lambda_{eff}$ | Effective thermal conductivity of the soil | $\mathrm{W\,m^{-1}\,K^{-1}}$ |
| $\rho_{ice}$ | Density of ice | $\mathrm{kg\,m^{-3}}$ |
| $\rho_L$ | Density of soil liquid water | $\mathrm{kg\,m^{-3}}$ |
| $\rho_{soil}$ | Bulk soil density | $\mathrm{kg\,m^{-3}}$ |
| $\rho_s$ | Density of solids | $\mathrm{kg\,m^{-3}}$ |
| $\rho_V$ | Density of water vapor | $\mathrm{kg\,m^{-3}}$ |
| $\theta$ | Volumetric water content | $\mathrm{m^3\,m^{-3}}$ |

| | | |
|---|---|---|
| $\theta_{ice}$ | Soil ice volumetric water content | $m^3\ m^{-3}$ |
| $\theta_L$ | Soil liquid volumetric water content | $m^3\ m^{-3}$ |
| $\theta_r$ | Residual soil water content | $m^3\ m^{-3}$ |
| $\theta_s$ | Volumetric fraction of solids in the soil | $m^3\ m^{-3}$ |
| $\theta_{sat}$ | Saturated soil water content | $m^3\ m^{-3}$ |
| $\theta_V$ | Soil vapor volumetric water content | $m^3\ m^{-3}$ |

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

 **Tables and Figures**

**Table 1. Numerical experiments with various mass and energy transfer processes**

| Experiments | Soil Physical Processes | | Model Components |
| | Unfrozen period | Frozen period | |
| --- | --- | --- | --- |
| unCPLD | Independent water and heat transfer | Independent water and heat transfer; No ice effect on soil properties; No latent heat due to phase change; | T&C (Eqs. 1 & 3) |
| unCPLD-FT | Independent water and heat transfer | FT induced water and heat transfer coupling; Ice effect on soil properties; Latent heat due to phase change; | T&C-FT (Eqs. 1 & 4) |
| CPLD | Tightly coupled water and heat transfer | Tightly coupled water and heat transfer; Ice effect on soil properties; Latent heat due to phase change; Convective heat due to liquid/vapor flow. | T&C-STEMMUS (Eqs. 7 & 8) |

Note:

Independent water and heat transfer: Soil water and heat transfer process is independent.

 FT induced water and heat transfer coupling: Soil water and heat transfer process is coupled only during the freezing/thawing (FT) period. Soil water flow is affected by temperature only through the presence of soil ice content (the impedance effect).

Tightly coupled water and heat transfer: Soil water and heat transfer process is tightly coupled; vapor flow, which links the soil water and heat flow, is taken into account; thermal effect on water flow is considered (the hydraulic conductivity and matric potential is dependent on soil temperature; when soil freezes, the hydraulic conductivity is  reduced by the presence of soil ice, which is temperature dependent); the convective/advective heat due to liquid/vapor flow can be calculated.

Ice effect on soil properties: the explicit simulation of ice content and its effect on the hydraulic/thermal properties.

**Table 2. Monthly values of energy closure ratio derived from eddy covariance measured LE + H versus Rn and Rn-G$_0$, respectively (Dec. 2017-Aug. 2018). G$_0$, the ground heat flux, was estimated by CPLD model.**

| Energy closure ratio | Dec | Jan | Feb | Mar | Apr | May | Jun | Jul | Aug |
|---|---|---|---|---|---|---|---|---|---|
| (*LE+H*) vs *Rn* | 0.58 | 0.58 | 0.61 | 0.45 | 0.53 | 0.55 | 0.55 | 0.57 | 0.59 |
| (*LE+H*) vs (*Rn-G$_0$*) | 0.98 | 0.90 | 0.90 | 0.51 | 0.62 | 0.68 | 0.64 | 0.63 | 0.67 |


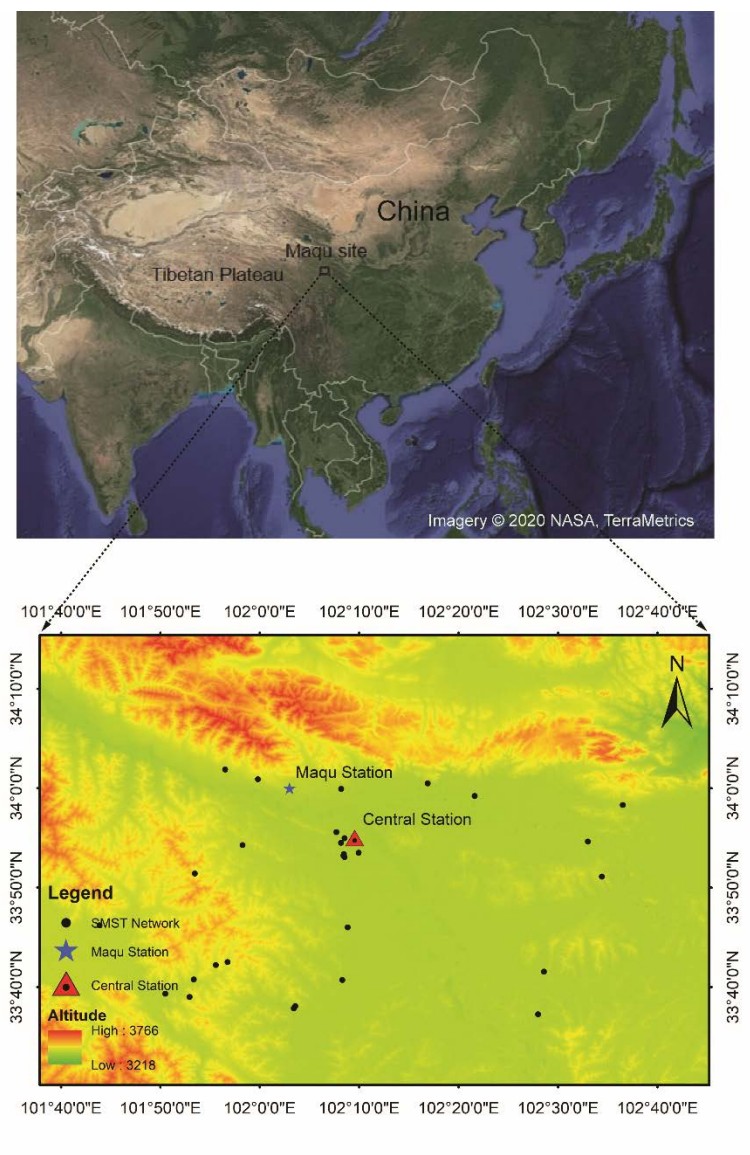

**Figure 1. Geographical location of Maqu soil moisture/temperature (SMST) monitoring network and the Centre station.**

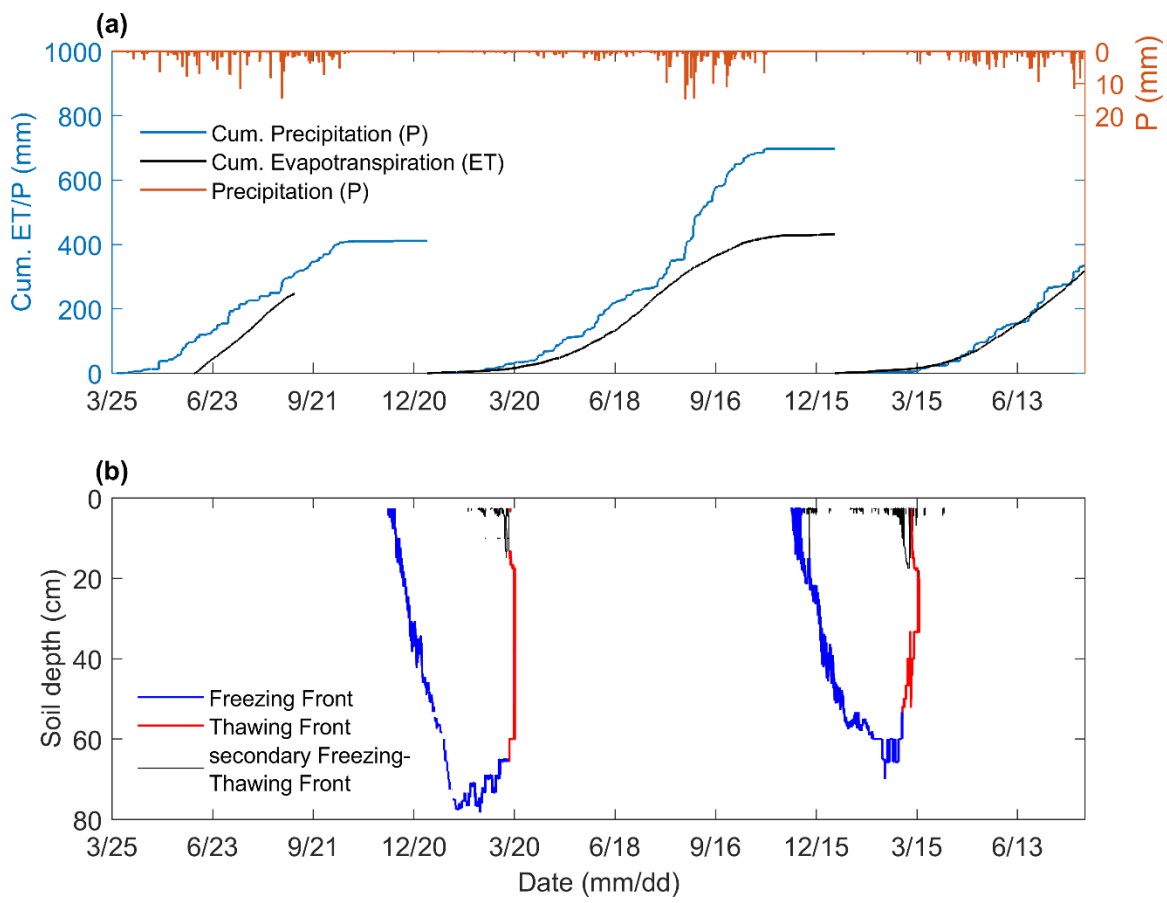


**Figure 2. (a) Observed cumulative precipitation (P) and evapotranspiration (ET) and (b) observed propagation of freezing/thawing front, with the blue, red, and black color for the primary propagation of freezing front and thawing front (FF & TF), and the secondary freezing-thawing front (sFTF) occurring at top soil layers, respectively, for the period 25 Mar. 2016- 12 Aug. 2018 at Maqu site.**


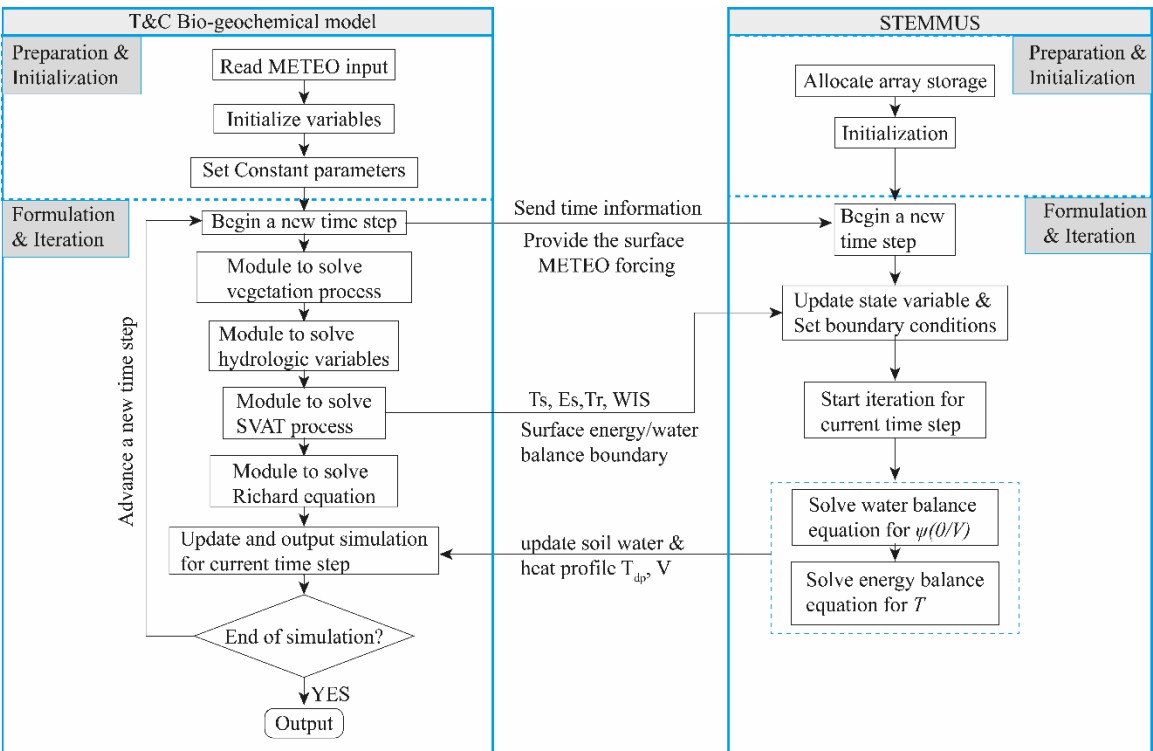

**Figure 3. Coupling procedure of STEMMUS and T&C model. METEO is the meteorology forcing, SVAT is acronym for the Soil-Vegetation-Atmosphere mass and heat Transfer. Ts, Es, Tr, WIS are the surface temperature, soil evaporation, plant transpiration, and incoming water flux to the soil, respectively. $T_{dp}$ and V are the soil profiles of temperature in °C and liquid water volume in each layer (mm).**


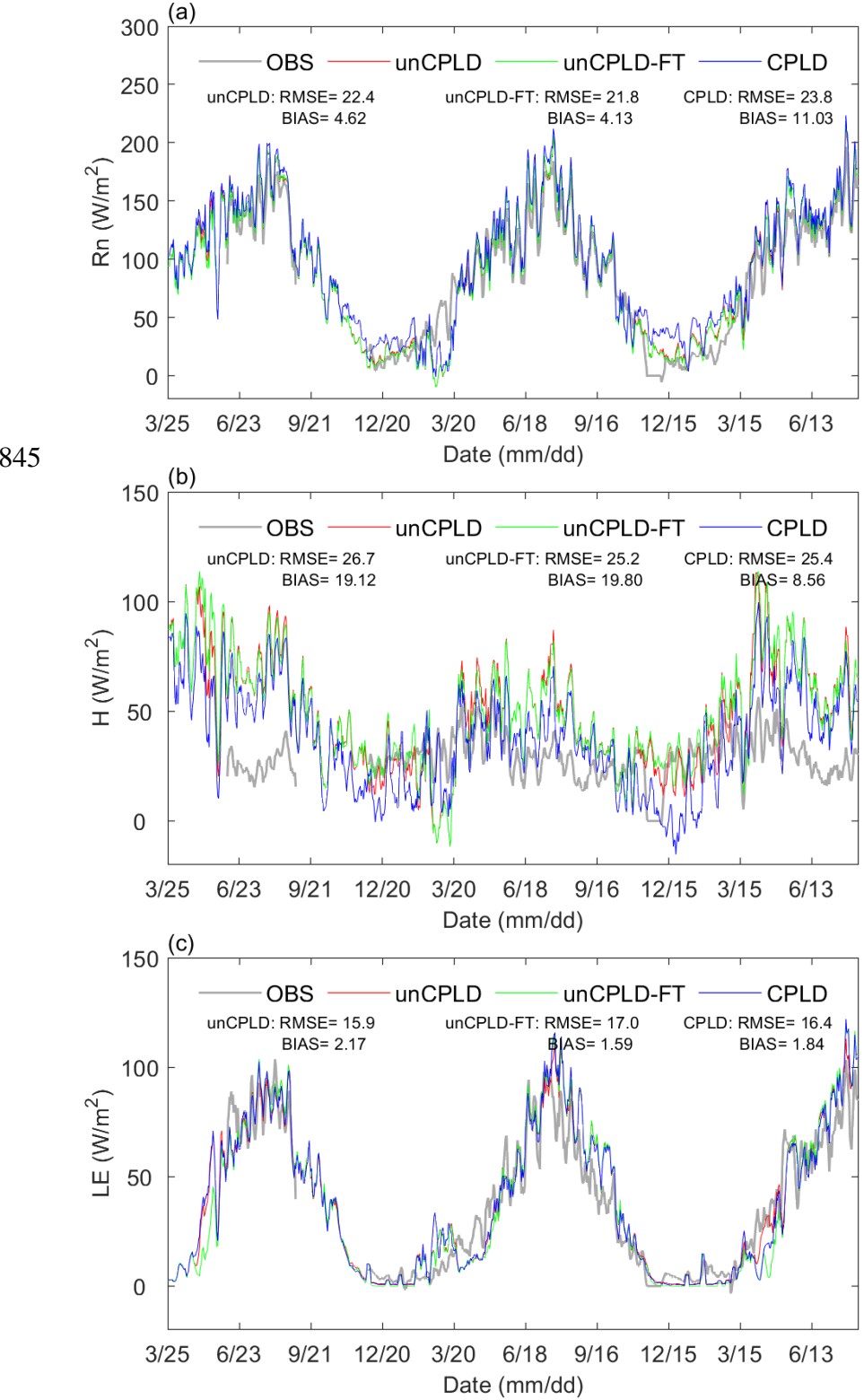


**Figure 4. Comparison of observed and simulated 5-day moving average dynamics of net radiation (Rn), latent heat flux (LE), and sensible heat flux (H) using the original (uncoupled) T&C (unCPLD), T&C with consideration of FT process (unCPLD-FT) and coupled T&C and STEMMUS (CPLD) model.**


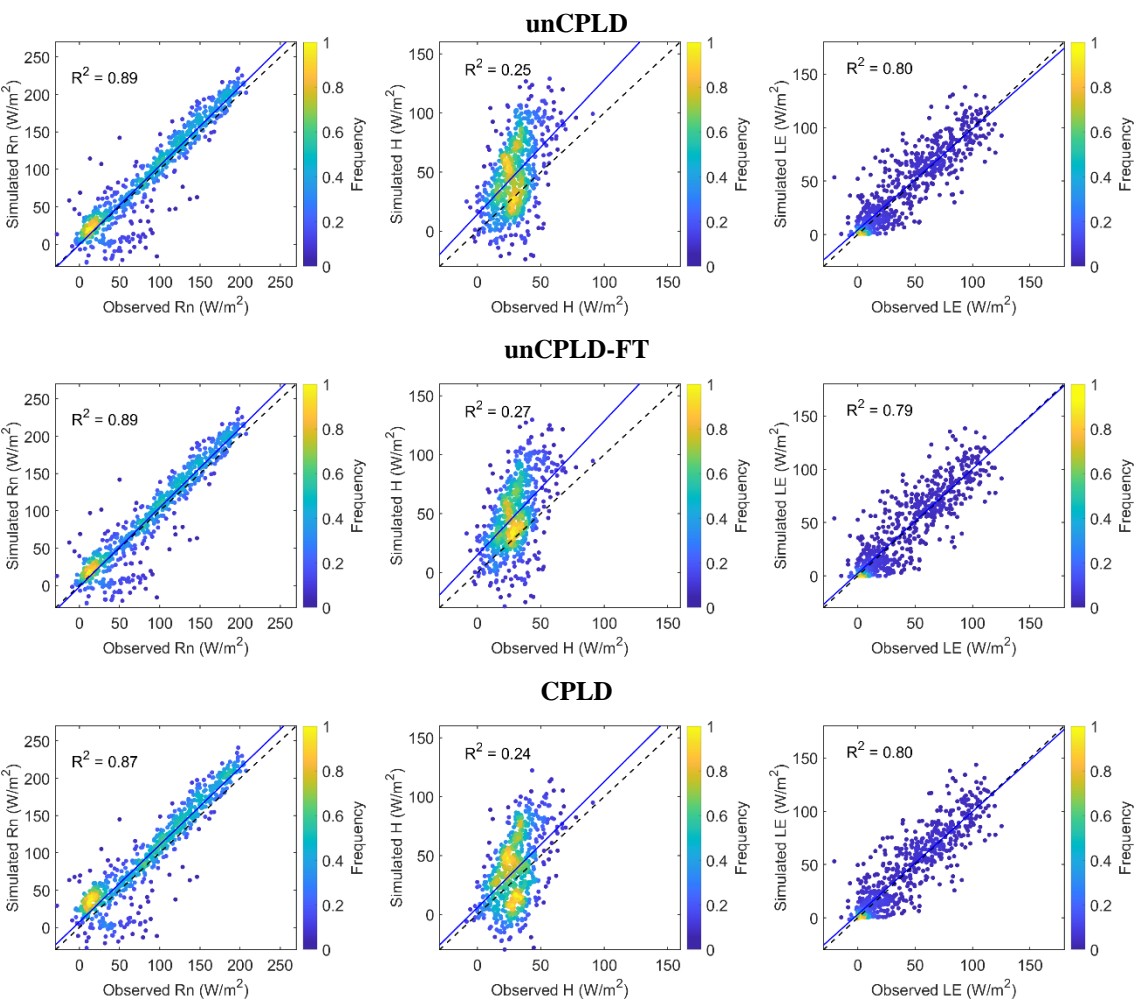

**Figure 5. Scatter plots of observed and model simulated daily average surface fluxes (net radiation: Rn, latent**
**heat: LE and sensible heat flux: H) using the original (uncoupled) T&C (unCPLD), T&C with consideration of FT process (unCPLD-FT) and coupled T&C and STEMMUS (CPLD) model, with the color indicating the frequency of surface flux values.**


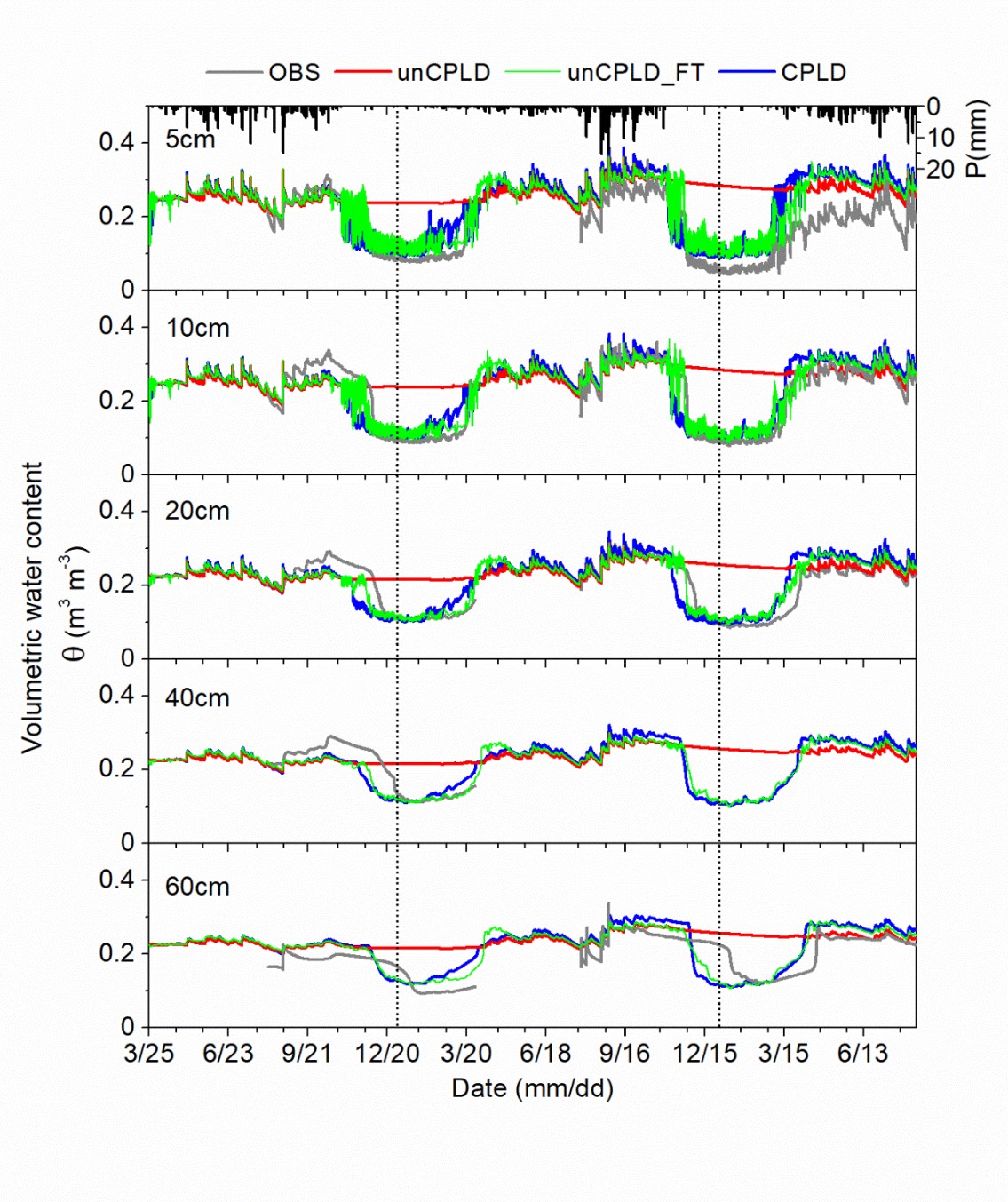

**Figure 6. Measured and estimated soil moisture at various soil layers using uncoupled T&C (unCPLD), uncoupled T&C with FT process (unCPLD-FT) and coupled T&C and STEMMUS (CPLD) model. Note that in unCPLD model, soil ice content is not explicitly considered, thus all the water remains in a liquid phase, which leads to a strong overestimation of winter soil water content in frozen soils.**


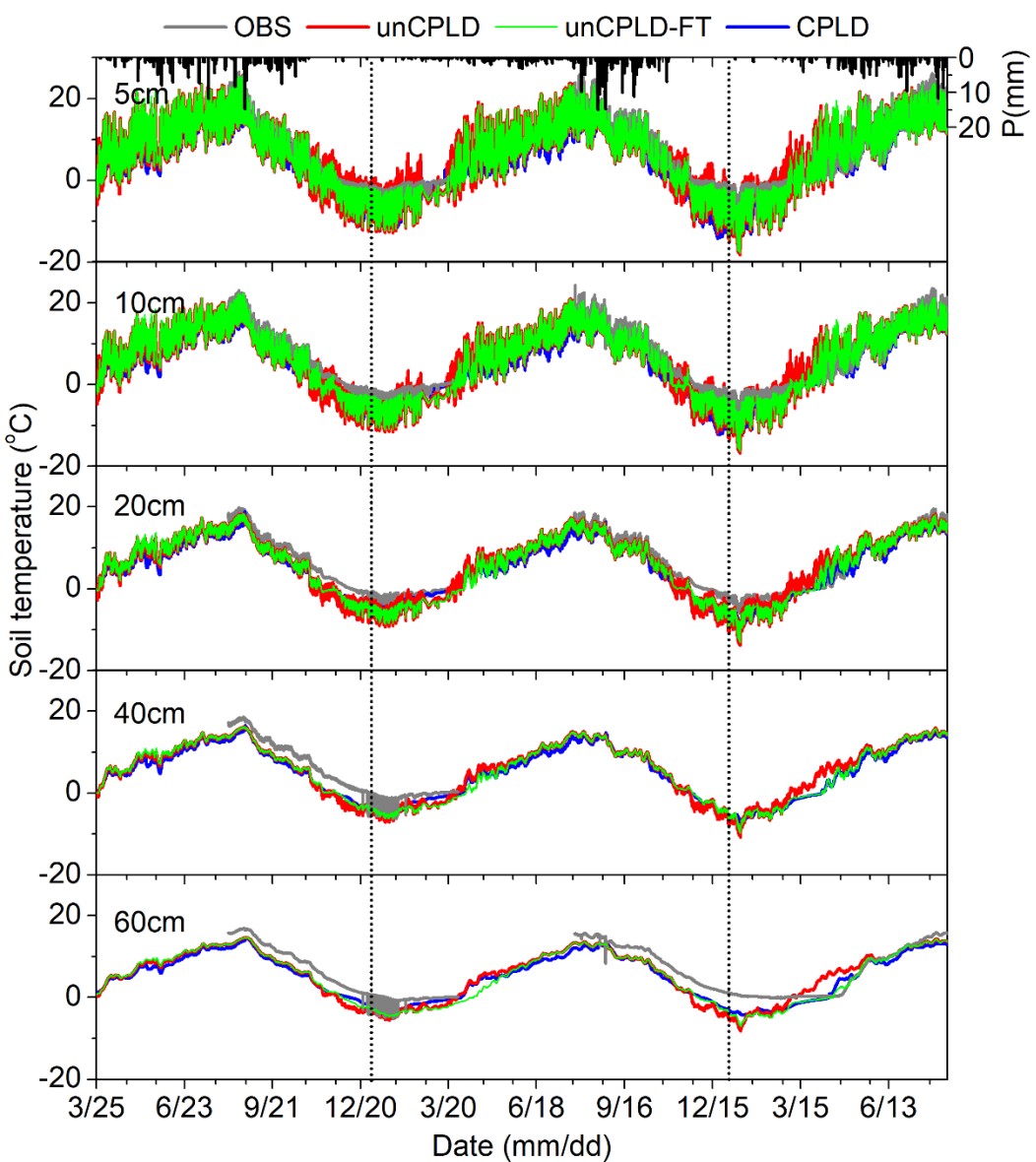

**Figure 7. Measured and simulated soil temperature at various soil layers using uncoupled T&C (unCPLD), T&C with FT process (unCPLD-FT) and coupled T&C and STEMMUS (CPLD) model.**



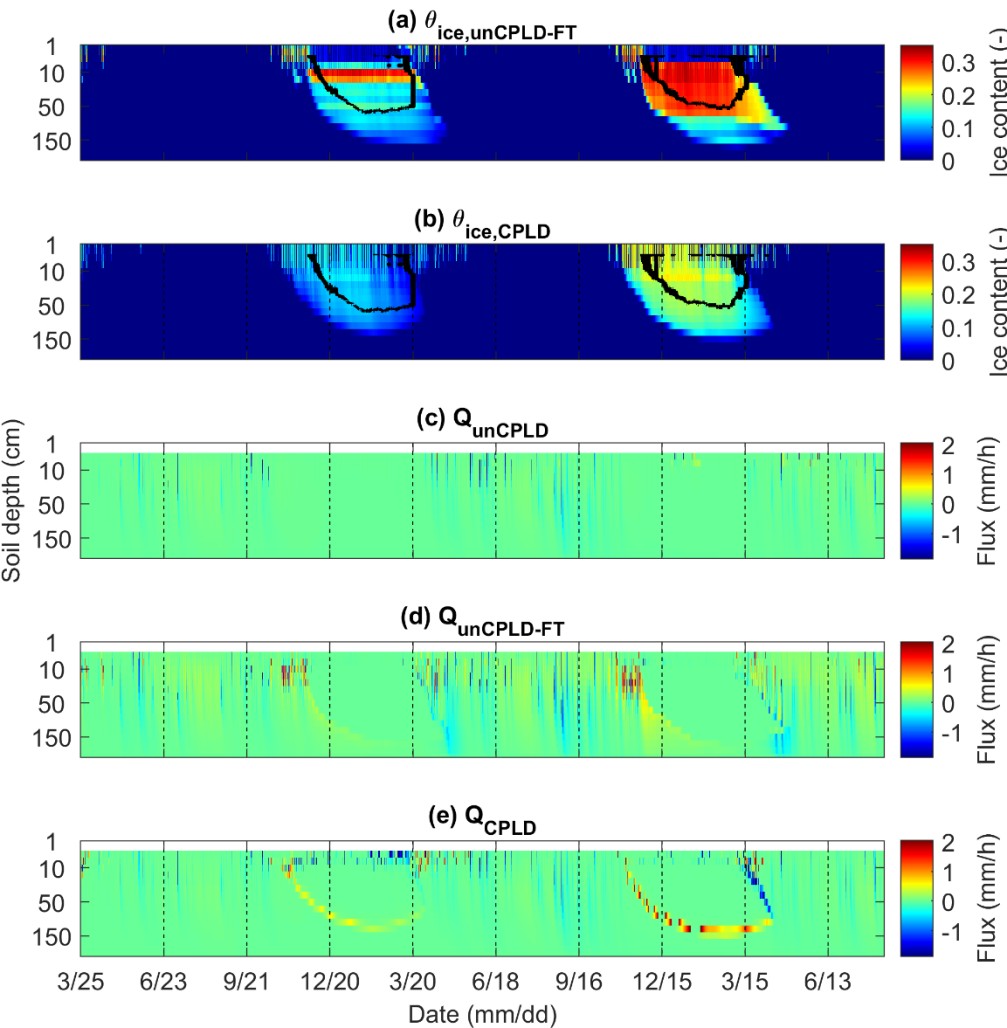

**Figure 8. Soil ice content from (a) unCPLD-FT and (b) CPLD model simulations with freezing front propagation derived from the measured soil temperature; and vertical water flux (positive value indicates upward water flow) from (c) unCPLD, (d) unCPLD-FT and (e) CPLD model simulations. Note that soil ice content is not represented in the unCPLD model and the fluxes of top 2 cm soil layers were not reported to highlight fluxes of the lower layers.**



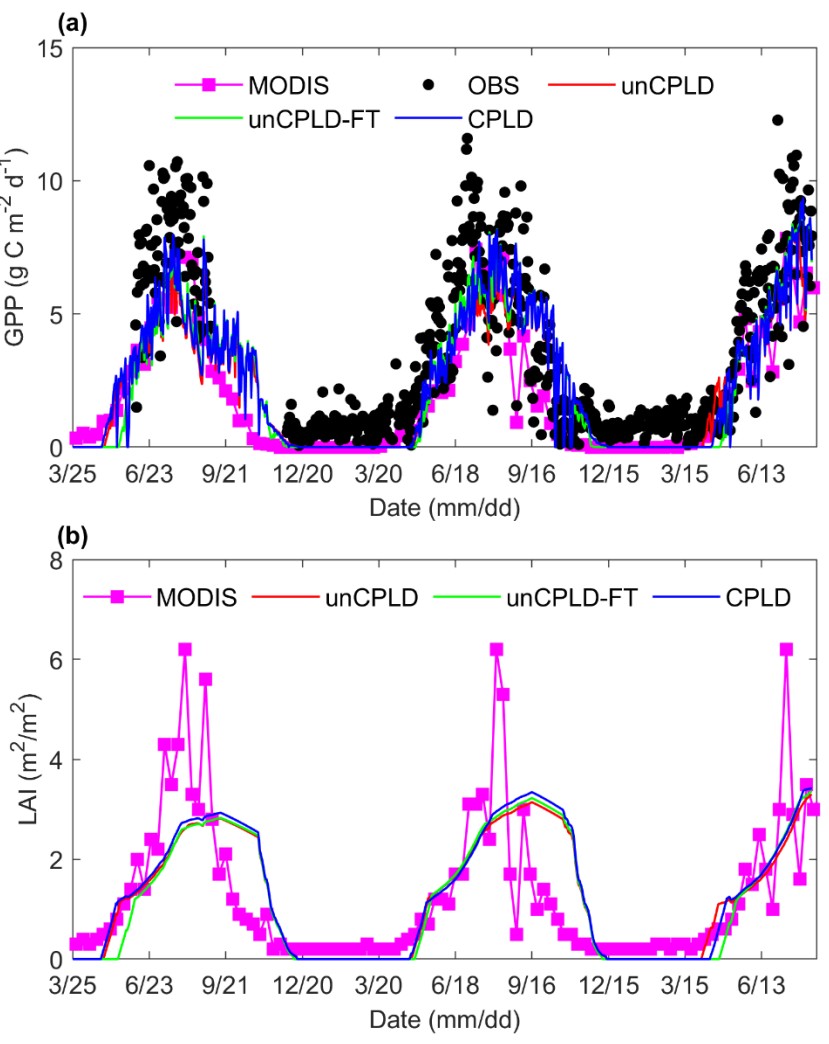

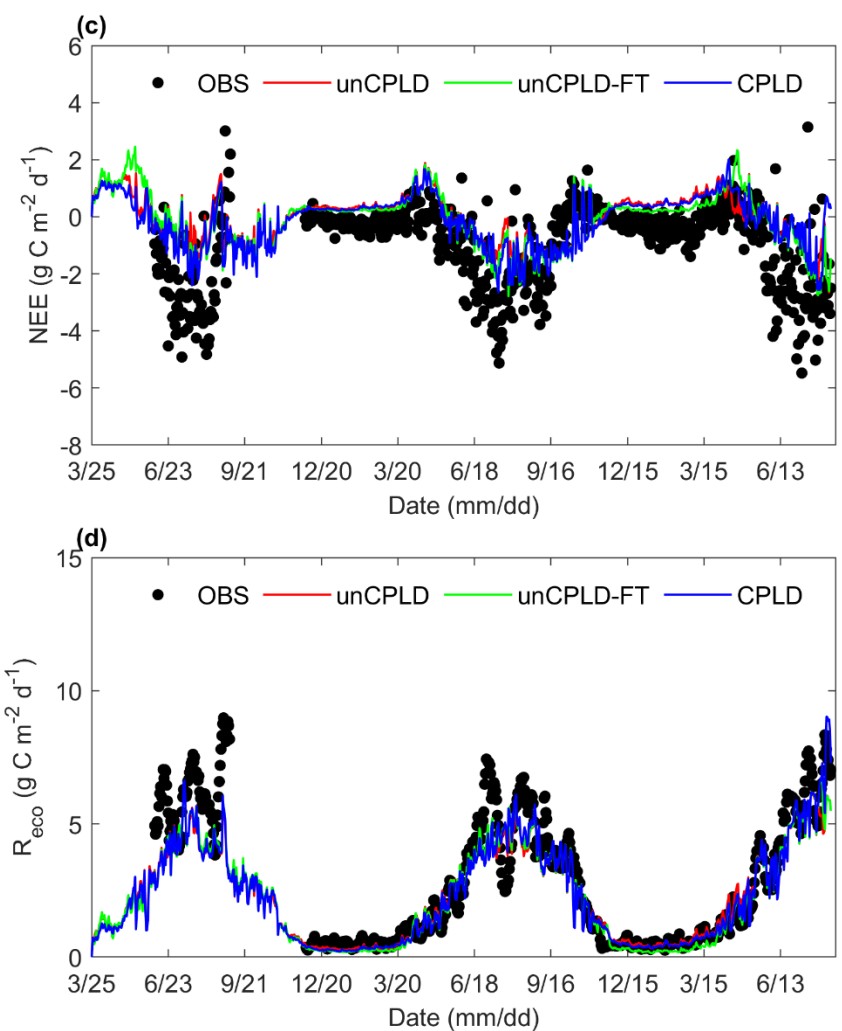

**Figure 9. Comparison of observations from Eddy Covariance (OBS) or MODIS remote sensing and simulated (a) Gross Primary Production (GPP), (b) Leaf Area Index (LAI), (c) Net Ecosystem Exchange (NEE), and (d) Ecosystem respiration ($R_{eco}$) using unCPLD, unCPLD-FT, and CPLD model. MODIS refers to the data from MODIS-GPP and MODIS-LAI products.**



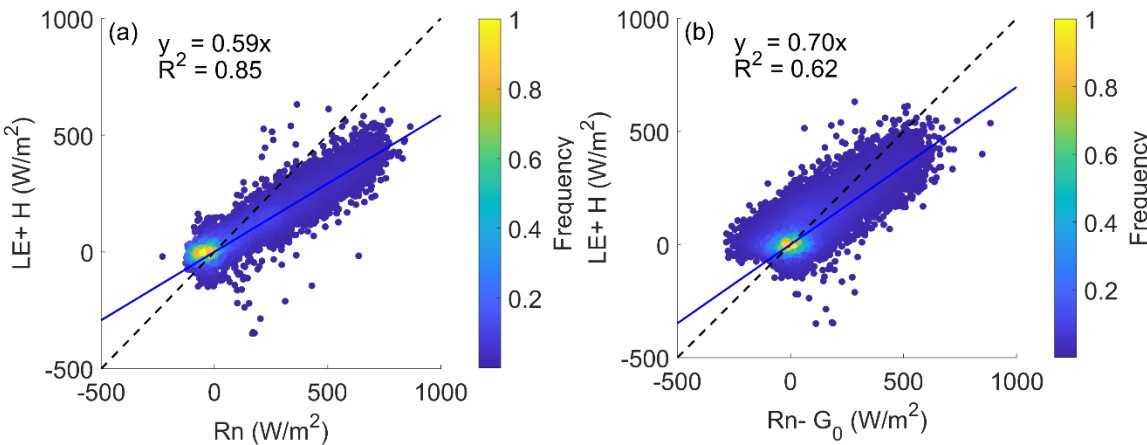

**Figure 10. Scatter plots of eddy covariance measured hourly values of LE + H versus (a) Rn and (b) Rn-G₀, with the color indicating the occurrence frequency of surface flux values. G₀, the ground heat flux, was estimated by the CPLD model.**


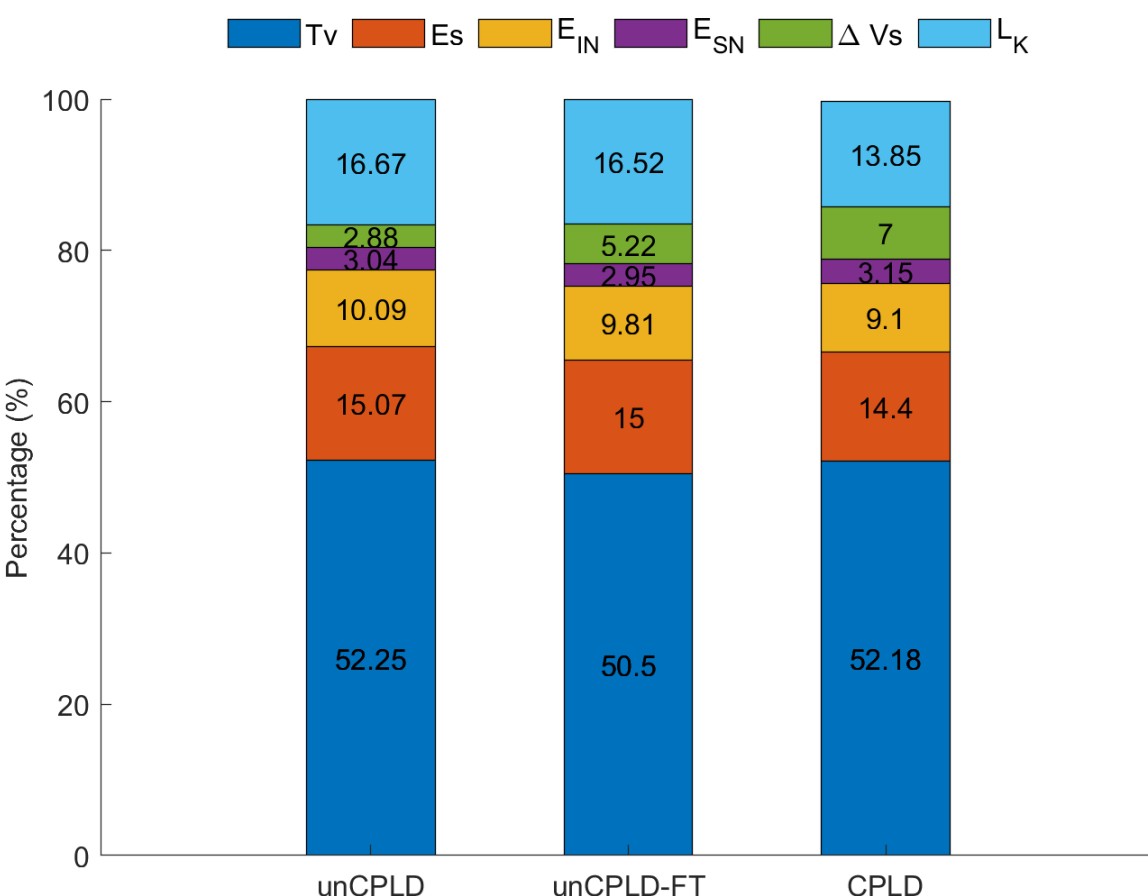

**Figure 11. Comparison of the relative ratios of different water budget components to precipitation during the whole simulation period produced by different model scenarios. Tv, transpiration; Es, surface evaporation; $E_{IN}$ and $E_{SN}$, evaporation from intercepted canopy water and snow cover; $\Delta$ Vs, changes in soil water storage; $L_K$, deep leakage water.**
