# Peer review of "The Role of Vadose Zone Physics in the Ecohydrological Response of a Tibetan Meadow to Freeze-Thaw Cycles"

_The Cryosphere, 2020_

## Referee Comment (RC1) · Anonymous Referee #1 · 17 Jun 2020

The manuscript by Yu et al. presents a study assessing the role of model complexity of soil physics processes on simulation of vegetation dynamics for a Tibetan meadow site. Three different model versions of the T&C model, with a gradual increase in the complexity of the freezing-thawing treatment, are compared: (1) the original T&C model, lacking soil freezing processes; (2) a modification of T&C in which an ice fraction and freezing/thawing are accounted for; and (3) a coupling of T&C to the soil physics model STEMMUS.

The model versions are parameterized and driven with data from a Tibetan meadow site to evaluate their performance. Differences between results with the three versions

of the model are small for most variables, generally smaller than the model-to-data difference, and the simulated differences are confined to parts of the year where freezing dynamics are likely to play a role. The analysis would benefit from a focus on specific periods of the year where differences arise.

This manuscript is potentially of interest to the readers of The Cryosphere and contains an interesting discussion on the role of freezing-thawing for ecosystem processes in high latitude and high altitude environments, and the importance of model complexity in ecohydrological models. However, in its current form, it contains many inaccuracies in the presentation of the results and ambiguity in the presentation of the model versions, simulation setup and the results, which make it hard to judge the models' qualities. I cannot recommend publication given the current state of the manuscript, but with substantial modifications in the texts and the presentation of the results, I expect that this could become suitable for publication in The Cryosphere. I will discuss my main objections per section below:

Title and introduction

The introduction discusses model improvement in very broad terms, but it does not give a rationale for focusing on soil freezing processes specifically in this study. I would suggest using the introduction instead for a more in-depth discussion of the role of freezing-thawing processes for ecosystem dynamics and the limitations of current models in representing these.

Specific remarks

- Title: The current title is grammatically incorrect. I would suggest removing the question mark at the end of the sentence. Alternatively, "affects" can be replaced by "does affect" (i.e., How does vadose zone mass and energy transfer physics affect the . . .")

- Introduction: The first part of the introduction (L. 33-43) is not very informative for the problem assessed in this manuscript: It describes in general terms the gradual

improvements that have been undertaken in many different types of models, without a clear focus on the research of

- L. 16: consider inserting "those relevant for" after "parameterizations"

- L. 23: "The difference among various complexity...": This sentence is unclear; the meaning of "among various complexity" needs to be specified.

- L. 26: Remove comma; also, I think that "in ecosystem functioning" should read "for ecosystem functioning".

- L. 58: "there are divergences". Please explain these divergences.

- L. 69: "The limited knowledge of including or not complex vadose zone processes..." Please clarify this sentence.

Methods

The model descriptions in the methods are somewhat unstructured and in part difficult to follow. I see the value of presenting the equations, but I would suggest to introduce the three model setups in the beginning of the methods section, and to describe the processes and equations per setup. For the numerical experiments (section 2.6), please specify which driving variables were used and at which temporal resolution, and how the initial state of the model was determined – without understanding the driving variables of the model, it is hard to evaluate the performance.

Specific remarks

- L. 101: Please specify whether you use remote sensing data for one pixel only, or whether you use it for a spatial analysis, and what it hence is "representative" (L. 102) for.

- Fig. 1b: It is hard to interpret the freezing front data, in particular for winter 2017-18 because of the missing data. Is it possible to mark the times and depths of missing data? Also, smaller symbols in the figure would probably allow to differentiate the

dynamics at the surface better.

- For the subscripts used in the equations, please separate the i used for layering (Eq. 2) from the i used for ice (Eq. 3, L. 196, Eq. 5).

- Eq. 4: Please check this equation: the left-hand side is mass-based, and the right-hand side volume-based. I assume that soil density should be added to the equation.

- L. 235: Convergence of which variables, and which criterion is used to determine if convergence has been achieved?

Results

The energy fluxes displayed in Fig. 3 and 4 are too large, probably by a constant factor (the seasonal dynamics look fine), resulting in fluxes that exceed theoretical limits set by incoming radiation – please check the averaging method. Numbers in Fig. 9 seem more realistic.

In general, the analysis focuses on the entire period of simulation. This is fine for a general overview, but differences between the three model versions tend to be small for most of the period. I would recommend focusing on specific times of the year where the three model versions deviate, and discuss the abilities of the three models for these periods specifically. Also, the authors could consider displaying differences from observations rather than absolute amounts, to make differences between model versions more visible. At the moment, the main conclusion that one draws as reader is that the choice of model version does not matter too much, whereas the differences between the model versions may well provide important insights e.g. in representing fluxes during freezing times.

Specific remarks:

- Can the authors comment on the differences in the models' abilities to capture LE and H? The model is doing a very good job in capturing LE, but variations in H are poorly captured. Regarding the "overall performance . . . in terms of turbulent flux simulation", I think this difference between LE and H should be noted. Also, the models simulate consistently a large difference in H between the summer from 2017 and those from 2016 and 2018, whereas observations indicate less variations between summers. What is the reason for the simulated differences between 2016 and 2018 on the one hand and 2017 on the other?

- L. 304 and Fig. 7ab: What causes the pronounced difference in simulated ice content between the two model versions, and is the band of high ice content in unCPLD-FT in the first winter season a model artefact or a real phenomenon? For comparing, it would be preferable to have the same colour scale for plots 7a and 7b.

- L. 326: What causes the difference in onset between unCPLD and CPLD on the one hand and unCPLD-FT on the other? How do the soil physics processes impact GPP?

Discussion and Conclusions

The discussion is generally fine and provides insights in how soil physics processes are expected to affect other parts of the ecohydrological system. It would be nice to see whether conclusions from the authors corroborate existing literature, and where they agree. The conclusion provides a balanced assessment of the advantages and disadvantages of enhanced model complexity for representing the dynamics.

- L. 361: Specify which slope is discussed here (it is clear from the figure, but hard to understand from the text).

General comments

The language would benefit from editing by a native speaker. Also, references should be checked carefully; references seem to be missing from the reference list (Fisher et al. 2014) or need to be specified (Yu et al. 2016a and 2016b in the reference list, but the text refers to Yu et al. 2016).

---

## Referee Comment (RC2) · Anonymous Referee #2 · 22 Jun 2020

"How vadose zone mass and energy transfer physics affects the ecohydrological dynamics of a Tibetan meadow?" by Yu et al. addresses the importance of including freezing and thawing processes and coupling water and energy balance equations of vadose zone models to simulate ecohydrological dynamics. The authors use data from a high-elevation meadow of the Tibetan plateau to evaluate the outcomes of their different simulations. From their results, the authors claim that adding model complexity improves the model estimates of soil temperature and moisture but may not be necessary to model vegetation dynamics in these regions.

The manuscript is generally well written and understandable. While the presentation

and interpretation of the results makes sense, it sometimes lacks clarifications and could benefit from a deeper analysis. I think the model, even fully coupled, does a poor job at reproducing ice content and thermal conditions of deeper layers. As pointed by the authors, some data collected to evaluate the different simulations seem wrong, while others are simply missing, which means it can be barely used to validate any of the simulations. However, the conclusions of this manuscript remain valid and of considerable interest and is valuable to anyone wondering about the importance of including dynamic freezing and thawing processes in their models. Overall, this manuscript has the potential to be an interesting and useful contribution to The Cryosphere, but some critical issues, mostly in the methodology section, need to be addressed before considering publication.

I also think this manuscript does not reach its full potential. It could use various level of complexity in their approach to include freezing and thawing processes. For example, the CPLD simulation could be divided into different simulations: one that only considers latent heat, one that only considers the effect of ice content on the hydraulic conductivity, etc. This would show modelers which component of the freezing/thawing processes is important to consider, and which one is potentially not. However, I acknowledge that this suggestion would involve substantial additional work, so I understand if the authors would rather not make this change.

Here are my detailed comments.

Introduction: Well written and introduce well the topic and the problematic. However, I think the authors could add a few lines about the efforts done so far to model freezing and thawing processes in coupled water & energy models for cold regions. There are numerous subsurface and surface models already doing that. They do not all necessarily simulate ecological dynamics, but some studies have provided useful information about the impact of neglecting freezing dynamics.

L56: Unclear what is meant by "changes of frozen ground". Maybe the authors meant

something like "variations in seasonally frozen ground thickness"?

L58: What are those divergences? Please provide a few examples.

L70: I am not sure "complexity" is the best word here. According to the authors, they are testing their models with or without freezing dynamics, and with or without water and energy coupling. As I understand it, there are no different "complexities" in the way frozen soil is represented or in the way the coupling is achieved.

Section 2.1: It is unclear until the reader reaches the results section if the experimental site is underlain by permafrost or not. Please add somewhere in this section that the site only has seasonally frozen ground.

L93: It is written here that the SMST profiles are measuring temperature and soil moisture at a depth of 80 cm. However, figures from the results section are never showing data at 80 cm, only at 60 cm (which is not listed in line 93). Please fix or clarify.

Figure 1b: If the figure is indeed showing both freezing and thawing fronts, I would recommend using different colors. It is unclear which zone is thawed and which one is frozen (mostly in 2017-2018). I also think the y-axis should not reach 100 cm. Based on the text, there is no sensor deeper than 80 cm. If I understood correctly, the graph is plotting information the authors do not have. Please fix or clarify in the text. Furthermore, is the data here interpolated from the 5 sensors described in line 93?

L124: Please provide more information about the missing data. When and where? This should also be shown in Figure 1b. For example, the authors can use a greyed area to show where data are missing. This is crucial considering the authors are using this data or comparison purposes in Figure 6.

Equation 1: Check units. Are the units of S really s-1 or rather kg m-3 s-1?

Equation 2: Undefined variables: dz, rH, rL, nc

Equation 4: Csoil is presented as the specific heat capacity of the bulk soil, but I think
it is the volumetric heat capacity. The units do not match otherwise. The same issue arises in equations 6, 7 and 8. Equation 8 is from Hansson et al. (2004), where Csoil is defined as the volumetric heat capacity.

Equation 8: Please define which phase is represented by $d\theta$. It should be liquid water.

Section 2.4.4: The authors describe in detail the equations of the two models but do not explain the critical components of the added freezing/thawing processes, except for latent heat. They do provide references but, considering that this is supposedly an important aspect of this manuscript, they should at least describe in more details the different equations used. For example, the calculation of the hydraulic conductivity for a frozen medium is quite important. The authors refer here to Hansson et al. (2004), which uses an impedance factor. This method is widely used, but the impedance factor is an arbitrary number that is likely to change based on the type of soil. The authors never write which impedance factor is used. I am also concerned that the authors are using the method from Dall'Amico et al. (2011) for the soil freezing characteristic curve and the method from Hansson et al. (2004) for the apparent heat capacity. These two papers both use a form of the Clausius-Clapeyron equation with the van Genuchten model, but with different approaches. I could be wrong but, depending on how they have been used, these two methods may not be compatible with each other. More information is required here to make sure the freezing model used by the authors is valid.

Section 2.6: I think this section lacks some important clarifications. First, I think the differences between the models are more complicated than coupled/uncoupled. The vadose zone equations of T&C model are not coupled, because the water and heat equations are independent from each other. On the other end, the heat equation of STEMMUS needs to be coupled to the mass transfer equation because of the consideration of different processes or constituents such as heat advection. However, when adding the freezing/thawing processes, the heat and water equations of T&C becomes somehow coupled (at least one-way) due to the temperature dependency of the hydraulic conductivity. It is unclear if the authors used the names unCPLD, unCPLD-FT and CPLD for their models to characterize the way the heat and water equations are solved or to characterize how the two components (T&C and STEMMUS) are used. If it is the former, I suggest that they authors rename their simulations as T&C, T&C-FT and T&C-STEMMUS. In any case, the coupling characteristics of the different simulations should be further explained. Secondly, it is unclear which processes/parameters are considered in each simulation. For example, the authors state in the discussion that "unCPLD-FT simulation accounting for soil-freezing in a simplified way in comparison to STEMMUS (e.g., the CPLD simulation)" (line 418-419). However, to my knowledge, the difference in the way STEMMUS accounts for soil-freezing processes is never explained. A paragraph describing the different processes that each simulation is accounting for is necessary in this section. This can also take the form of a table.

Figure 3: The grey line is hard to distinguish. I suggest making it slightly darker or thicker.

Figures 5 and 6: It looks like deeper soil moisture and temperature is not well reproduced by any of the models, even though it is not discussed in the text. This has the consequence of poorly representing ice content (Figure 7), mostly in 2017-2018. It is understandable considering the model has not been calibrated and the goal of this manuscript, which focuses more on the growing season, is not necessarily to validate the models. However, I think this poor fit with field measurement should be discussed in the text. There are many reasons that could explain this, such as the presence of heterogeneity in the soil or of freezing-point depression due to increased salinity.

L303: I think they authors meant "unCPLD-FT" instead of "unCPLD".

L307-309: I do not agree that the CPLD model shows a good match with field measurements of ice content, at least not as currently showed in winter 2017-2018.

Figure 8: Please define acronyms in the caption (e.g., "(a) Gross Primary Production" instead of "(a) GPP")

Figure 9: The authors use $R^2$ here and R in Figure 8. I suggest them to choose one and be consistent.

Figure 10: Please define acronyms in the figure caption.

L381-382: Confusing sentence. Should we compare to unCPLD or unCPLD-FT?

L383-384: What could explain cooler late winter temperatures in unCPLD-FT? Latent heat slowing down the thawing? Lower bulk heat capacity of frozen soil? Please provide hypotheses.

L396-400: This analysis could be improved. The coupling is not the only difference between CPLD and unCPLD-FT. STEMMUS is simulating some subsurface processes that T&C does not (e.g., heat advection, air flow, vapor flow). I recommend providing a more detailed analysis then simply justifying the differences by the coupling. Also, how is ice content and hydraulic conductivity being simulated differently in CPLD than in unCPLD-FT?

L407-410: There are two requirements to experience heat advection: water flow and difference in temperature. While the former is shown in Figure 7, there is no evidence shown for the latter. It would be interesting if the authors could provide some evidence (can be with numbers or words) that heat advection (or convective heat) is mostly relevant during the frozen period.

L413: I think Figure 8 should be referred here instead of Figure 9.

---

## Author Comment (AC1) · 31 Jul 2020

We thank the editor for the time and effort in facilitating the reviewing process and the reviewers for their constructive comments. Please see our point by point response as follows. The reviewers' comments are in **black fonts** indexed by numbers. Our response is in blue fonts and **red fonts** are the updates in the manuscript.

General Comment: The manuscript by Yu et al. presents a study assessing the role of model complexity of soil physics processes on simulation of vegetation dynamics for a Tibetan meadow site. Three different model versions of the T&C model, with a gradual increase in the complexity of the freezing-thawing treatment, are compared: (1) the

original T&C model, lacking soil freezing processes; (2) a modification of T&C in which an ice fraction and freezing/thawing are accounted for; and (3) a coupling of T&C to the soil physics model STEMMUS. The model versions are parameterized and driven with data from a Tibetan meadow site to evaluate their performance. Differences between results with the three versions of the model are small for most variables, generally smaller than the model-to-data difference, and the simulated differences are confined to parts of the year where freezing dynamics are likely to play a role. The analysis would benefit from a focus on specific periods of the year where differences arise.

This manuscript is potentially of interest to the readers of The Cryosphere and contains an interesting discussion on the role of freezing-thawing for ecosystem processes in high latitude and high altitude environments, and the importance of model complexity in ecohydrological models. However, in its current form, it contains many inaccuracies in the presentation of the results and ambiguity in the presentation of the model versions, simulation setup and the results, which make it hard to judge the models' qualities. I cannot recommend publication given the current state of the manuscript, but with substantial modifications in the texts and the presentation of the results, I expect that this could become suitable for publication in The Cryosphere.

Response: Thanks a lot for your insightful comments. We rearranged the manuscript structure to make it more readable. The experimental site (and its multi-component measurements) was introduced in Section 2. The STEMMUS and T&C models (and their coupling), as well as numerical experiment designs (and key governing equations) were introduced in Section 3. Equations for different model versions were clearly presented in Sect. 3.1-3.4. Section 4 detailed results and discussions on the role of different vadose zone physics in the ecohydrological response to freeze-thaw cycles. Section 5 summarized the potential influential pathways of different vadose zone physics, and the study was concluded in Section 6. We added Table 1 to clarify the difference among three models (see Sect. 3.5). The constitutive equations regarding unfrozen water content, ice effect on hydraulic conductivity, the temperature depen-
dence of water flow, water vapor density were given in Supplement S1. We added figures in supplement materials to present the surface energy fluxes simulations during the non-frozen and frozen period, respectively. The relevant text was added to the manuscript and supplement Section S2 to explain the differences among the three models. We added Figure 1 to illustrate the geophysical location of Maqu soil moisture and soil temperature (SMST) monitoring network, indicating the central experimental site where our data were collected. Thus the figure numbers in the revised manuscript were all updated correspondingly.

1. The introduction discusses model improvement in very broad terms, but it does not give a rationale for focusing on soil freezing processes specifically in this study. I would suggest using the introduction instead for a more in-depth discussion of the role of freezing-thawing processes for ecosystem dynamics and the limitations of current models in representing these.

Response: Thank you very much. We updated the introduction part adding more focus on the role of the freezing-thawing process for ecosystem dynamics and we discussed the limitation of current models in representing these processes.

2. - Title: The current title is grammatically incorrect. I would suggest removing the question mark at the end of the sentence. Alternatively, "affects" can be replaced by "does affect" (i.e., How does vadose zone mass and energy transfer physics affect the :::")

Response: Thanks a lot. We removed the question mark and rephrased the title as "On the Role of Vadose Zone Physics in the Ecohydrological Response of a Tibetan Meadow to Freeze-Thaw Cycles "

3. – Introduction: The first part of the introduction (L. 33-43) is not very informative for the problem assessed in this manuscript: It describes in general terms the gradual improvements that have been undertaken in many different types of models, without a clear focus on the research

TCD
Response: Thanks a lot for your insightful comments. We modified the first part of the introduction and focused on the response of ecosystem dynamics in cold regions, and on the ongoing modeling efforts with consideration of vegetation and freeze/thaw process. Then we further stressed the relevance to consider the coupling water and heat physics in cold regions.

Changes in the manuscript: "Recent climatic changes have accelerated frozen soil dynamics in cold regions, as for instance favoring permafrost thawing and degradation of frozen soils (Cheng and Wu, 2007;Hinzman et al., 2013;Peng et al., 2017;Yao et al., 2019;Zhao et al., 2019). In consequence of these changes, vegetation cover and phenology, land surface water and energy balances, subsurface soil hydrothermal regimes, water flow pathways were reported to be affected (Campbell and Laudon, 2019;Gao et al., 2018;Schuur et al., 2015;Walvoord and Kurylyk, 2016;Wang et al., 2012). Understanding how ecosystem functioning interacts with changing environmental conditions is a crucial yet challenging problem of Earth system research for high latitude/altitude regions and deserves further attention.

Land surface models, terrestrial biosphere models, ecohydrology models, and hydrological models have been widely utilized to enhance our knowledge in terms of land surface processes, ecohydrological processes (Fatichi et al., 2016a;Fisher et al., 2014), and freezing/thawing process (Cuntz and Haverd, 2018;Druel et al., 2019;Ekici et al., 2014;Wang and Yang, 2018;Wang et al., 2017b). For instance, Zhuang et al. (2001) incorporated a permafrost model into a large scale ecosystem model to investigate soil thermal temporal dynamics. Zhang et al. (2018) investigated the long term highlatitude Arctic tundra ecosystem response to the interannual variations of climate with the process based CoupModel. The LPJ-WHy model, with consideration of permafrost dynamics and peatland, was used to analyze land surface processes by Wania et al., (2009). Lyu and Zhuang (2018) coupled a soil thermal model with the Terrestrial Ecosystem Model (TEM) to explore snowpack effects on soil thermal and carbon dynamics of the Arctic ecosystem under different climate scenarios. "

**TCD**
4. - L. 16: consider inserting "those relevant for" after "parameterizations" Response: We rephrased the text here. "The physical representation is increased from T&C without, and with the explicit consideration of ice effect, to T&C coupling with STEMMUS enabling the simultaneous mass and energy transfer in the soil system (liquid, vapor, ice)."

5. - L. 23: "The difference among various complexity: : :": This sentence is unclear; the meaning of "among various complexity" needs to be specified.

**Response:** We modified this sentence as "The physical representation is increased from T&C without, and with the explicit consideration of ice effect, to T&C coupling with STEMMUS enabling the simultaneous mass and energy transfer in the soil system (liquid, vapor, ice)."

6. - L. 26: Remove comma; also, I think that "in ecosystem functioning" should read "for ecosystem functioning".

Response: We have removed the comma and replaced "in ecosystem functioning" with "for ecosystem functioning".

7. - L. 58: "there are divergences". Please explain these divergences.

Response: We explain the divergences in the context as "In response to climate warming, the degradation of frozen ground can positively affect the vegetation growth in Tibetan Plateau mountainous region (Qin et al., 2016), but it can also lead to degradation of grasslands (Cheng and Wu, 2007), depending on soil hydrothermal regimes and climate conditions (Qin et al., 2016;Wang et al., 2016)."

8. - L. 69: "The limited knowledge of including or not complex vadose zone processes: : " Please clarify this sentence.

Response: Here the complex vadose zone processes is referring to the "explicit consideration of ice effect, water and heat coupling". We want to say that there has not been too much effort investigating the role of the increasing complexity of soil physiTCD
cal processes in cold region ecosystems. we rephrased it as "The inclusion or not of different soil physical processes, i.e., explicit considering ice effect and tightly coupled water and heat transfer, in such environment frames the scope here."

9. Methods: The model descriptions in the methods are somewhat unstructured and in part difficult to follow. I see the value of presenting the equations, but I would suggest to introduce the three model setups in the beginning of the methods section, and to describe the processes and equations per setup. For the numerical experiments (section 2.6), please specify which driving variables were used and at which temporal resolution, and how the initial state of the model was determined – without understanding the driving variables of the model, it is hard to evaluate the performance.

Response: The method part regarding the model descriptions has been modified accordingly. We first present the soil physical processes used in T&C, T&C-FT, and STEMMUS model in Section 3.1-3.4. Then the coupling T&C and STEMMUS procedure is introduced in Section 3.4, followed by the design of numerical experiments in Section 3.5. The description of driving variables was added in Section 3.5 as "Hourly meteorological forcing (including downwelling solar radiation, precipitation, air temperature, relative humidity, wind speed, air pressure) was utilized to drive the models. For the adaptive time step of STEMMUS simulation, linear interpolation between two adjacent hourly meteorological measurements was used to generate the required second values."

10. - L. 101: Please specify whether you use remote sensing data for one pixel only, or whether you use it for a spatial analysis, and what it hence is "representative" (L. 102) for.

Response: As the lack of in situ measurements of time series of vegetation dynamics, here we intended to use remote sensing data (MCD15A3H and MOD17A2H) as the auxiliary data for the vegetation dynamics of the in situ site (corresponding to the central experimental site  $(33^{\circ}54'59"N, 102^{\circ}09'32", elevation: 3430m)$ ). One pixel data
corresponding to the study location was employed here in terms of the spatial scale. We rephrased the "representative" with "We downloaded MCD15A3H (Myneni et al., 2015) and MOD17A2H (Running et al., 2015) products for this site as the auxiliary vegetation dynamics data...".

11. - Fig. 1b: It is hard to interpret the freezing front data, in particular for winter 2017-18 because of the missing data. Is it possible to mark the times and depths of missing data? Also, smaller symbols in the figure would probably allow to differentiate the dynamics at the surface better.

Response: Figure 2b (original Figure 1b) was replotted accordingly. Here we generated the dynamics of freezing/thawing front by interpolating the measured soil temperature data at soil depths of 2.5, 5, 10, 20, 60, and 100cm, neglecting 40cm (at which depth the data is missing for 2017-18 winter). The missing data periods of soil moisture/temperature measurements were described in the text.

Changes in the manuscript: "Note that there are data gaps (25th Mar – 8th June, 2016; 29th Mar – 27th July, 2017, extended to 12th Aug, 2018 for 40 cm) due to the malfunction of instruments and the difficulty to maintain the network under harsh environment."

12. - For the subscripts used in the equations, please separate the i used for layering (Eq. 2) from the i used for ice (Eq. 3, L. 196, Eq. 5).

Response: We made modifications as: 'i ' is specifically used for layering and 'ice' is used for soil ice (changes can be found as Eq. 4, Eq. 6, L. 233, Eq. 7, Eq. 8).

13. - Eq. 4: Please check this equation: the left-hand side is mass-based, and the righthand side volume-based. I assume that soil density should be added to the equation.

Response: Many thanks for pointing it out. Eq. 4 (now Eq. 3): We added the soil density term ( $\rho_{soil}$ ) in Eq. 3, making the left-hand side consistent with the right-hand side both as volume-based terms. In addition, such modifications are made in Eq. 6 &

TCD
8.

14. - L. 235: Convergence of which variables, and which criterion is used to determine if convergence has been achieved?

Response: The iteration solution is used in STEMMUS to solve the soil moisture and temperature states. The convergency criteria are both set as 0.001 for soil matric potential and soil temperature.

Furthermore, the maximum desirable change of soil moisture and soil temperature within one step was set as  $0.02 \ cm^3 cm^{-3}$  and  $2 \ ^{\circ}C$ , respectively, to prevent too large change in state variables that may cause numerical instabilities. If the changes between two adjacent soil moisture/temperature states are less than the maximum desirable change, then STEMMUS continues without changing time step. Otherwise, STEMMUS will adjust the time step and repeat the current time step. During the freezing/thawing transition periods, we added the additional constraint on the time step to keep the smooth change of soil energy content. By decreasing the time step, the soil temperatures from two adjacent iterations were ensured either greater (smaller) than or equal to the freezing temperature (i.e., heating, cooling or zero-curtain effect).

Changes in the manuscript: "After convergence is achieved in the soil module (i.e., convergence criteria is set as 0.001 for both soil matric potential [in cm] and soil temperature [in  $^{\circ}C$ ])"

15. The energy fluxes displayed in Fig. 3 and 4 are too large, probably by a constant factor (the seasonal dynamics look fine), resulting in fluxes that exceed theoretical limits set by incoming radiation – please check the averaging method. Numbers in Fig. 9 seem more realistic.

Response: Thanks a lot for pointing it out. In the original Fig. 3 and 4, we used wrong units summing values of hourly surface energy fluxes (ranging from -500 to 1000  $W/m^2$  for net radiation) into daily values. For the updated Fig. 4 and 5, we corrected the mistake and we properly averaged the surface energy fluxes at the daily
time scale then presented the 5-day moving average values of surface energy fluxes (ranging from -20 to 200  $W/m^2$  for net radiation). For Fig. 10 (original Figure 9), the hourly values of surface energy fluxes, plotted as the scattered figures between the observed and model simulated values, were used to indicate the energy balance closure problem. We made some modifications in captains of Figures 4, 5 and Figure 10 to clearly indicate whether the hourly or daily average values are used.

16. In general, the analysis focuses on the entire period of simulation. This is fine for a general overview, but differences between the three model versions tend to be small for most of the period. I would recommend focusing on specific times of the year where the three model versions deviate, and discuss the abilities of the three models for these periods specifically. Also, the authors could consider displaying differences from observations rather than absolute amounts, to make differences between model versions more visible. At the moment, the main conclusion that one draws as reader is that the choice of model version does not matter too much, whereas the differences between the model versions may well provide important insights e.g. in representing fluxes during freezing times.

Response: Thanks for the suggestion. We added additional figures in the supplement materials to highlight differences in specific periods (see supplement Figures S1-3). We zoomed in a new figure (Fig. S1) in the frozen period and we showed the relative differences between observations and model simulations. We add an appropriate reference in the manuscript and supplement accordingly.

17. - Can the authors comment on the differences in the models' abilities to capture LE and H? The model is doing a very good job in capturing LE, but variations in H are poorly captured. Regarding the "overall performance : : : in terms of turbulent flux simulation", I think this difference between LE and H should be noted. Also, the models simulate consistently a large difference in H between the summer from 2017 and those from 2016 and 2018, whereas observations indicate less variations between summers. What is the reason for the simulated differences between 2016 and 2018 on the one

TCD
**hand and 2017 on the other?**

Response: It is difficult to attribute such a difference mostly to the model inaccuracy or mostly to the data inaccuracy. On one hand, the energy balance closure problem rises as the potential source of error and reason of discrepancy. The Eddy covariance observed LE and mostly H fluxes are underestimated when constrained by the surface energy closure during the summer periods (see Table 2). On the other hand, in T&C model, surface temperature is simplified and 'one single prognostic surface temperature' is computed, i.e. soil surface and vegetation surface temperature have the same value. The difference between the soil surface, vegetation surface and the assumed surface temperature can be a potential cause for such discrepancies in H.

In addition, soil moisture and temperature simulation fit changes corresponding to that of surface energy fluxes simulations (i.e., slightly better in 2017 than that in 2016 and 2018, see Fig. 6). The uncertainties in the precipitation measurements thus can be an additional potential reason for the simulated differences between 2016 & 2018 and 2017.

18. - L. 304 and Fig. 7ab: What causes the pronounced difference in simulated ice content between the two model versions, and is the band of high ice content in unCPLD-FT in the first winter season a model artefact or a real phenomenon? For comparing, it would be preferable to have the same colour scale for plots 7a and 7b.

Response: Soil ice content measurements are not easy to achieve in the field. It is hard to accurately assess the model performance regarding the soil ice content simulations. We simply rely on the freezing/thawing front propagation to validate the general spatiotemporal shape of soil ice content dynamics.

For the band of high ice content in unCPLD-FT in the first winter season, this is generated by cryosuction of liquid water in the upper soil layers. The freezing-induced water potential decrease moves the available liquid water towards the freezing front. The high accumulation of ice content indicates that unCPLD-FT model simulated a relatively strong cryosuction effect, probably mitigated in the fully coupled model by effects TCD
of water vapor transfer and thermal gradients, as well as different solutions in the parameterization of soil-freezing curves.

We now used the same colour scale for Figure 8a &b (original Figure 7).

Changes in the manuscript: "It is to note that compared to unCPLD-FT model, CPLD model presented a relatively lower presence of soil ice content, while its temporal dynamics was closer to the observed freezing/thawing front propagation. The difference between the two simulations can be attributed to the constraints imposed by the interdependence of liquid, ice and vapor in the soil pores that is considered only in CPLD model."

19. - L. 326: What causes the difference in onset between unCPLD and CPLD on the one hand and unCPLD-FT on the other? How do the soil physics processes impact GPP?

Response: The onset of vegetation depends on the soil temperature in the root zone averaged over the previous 30 days. For the first winter season, unCPLD-FT model simulated a prolonged freezing period than unCPLD and CPLD model. Thus, there is a delay in the vegetation onset date. For the second winter season, all three models produce similar vegetation onset dates. For the third winter season, more spread was detected among the three model simulations. unCPLD produced the earliest onset date while unCPLD-FT produced the latest onset date, with CPLD fell in between. The difference in soil physics processes alters the soil liquid water/temperature profile simulations and especially the strong cryosuction effect in the unCPLD-FT generates larger ice accumulation and a delay in the melting that leads to lower average temperatures. These processes affect the leaf onset date, i.e., the phenology of the grassland. Additionally, the changes in soil liquid water content can result in variations of water stress for the plants, thus they affect the photosynthetic assimilation rate and GPP. Differences in soil temperature profiles can also affect root respiration in generating additional small differences in GPP.

TCD
Changes in the manuscript: "The difference in the soil liquid water/temperature profile simulations between the CPLD and unCPLD models (as shown in Figures 6 & 7) resulted in differences in simulated vegetation dynamics, especially concerning the leaf onset date, which is affected by integrated winter soil temperatures. The unCPLD-FT model has a delay in the vegetation onset date when compared to other simulations, due to the significant cryosuction that prolongs freezing conditions and keep lower soil temperatures. This makes the unCPLD simulation having slightly shorter vegetation active season compared to the CPLD model simulations. The lower GPP in the unC-PLD simulations is instead related to a slightly enhanced water-stress induced by the different soil-moisture dynamics during the winter and summer season with a lower root zone moisture produced by the unCPLD model (Figure 6), which affects the plant photosynthesis and growth. Differences in soil temperature profiles can also affect root respiration in generating additional small differences in GPP. "

20. The discussion is generally fine and provides insights in how soil physics processes are expected to affect other parts of the ecohydrological system. It would be nice to see whether conclusions from the authors corroborate existing literature, and where they agree. The conclusion provides a balanced assessment of the advantages and disadvantages of enhanced model complexity for representing the dynamics.

Response: We thank the reviewer for this perspective. Additional references were added in the discussion part to expand it. For example, the 10yr CO2 fluxes observation in an alpine shrubland on the Qinghai-Tibetan Plateau by (Li et al. 2016) aligns well with this study as it indicated that the non-growing season soil temperature can exert important effects on the carbon flux dynamics, as it can enhance the vegetation activities and prolongs the growing season.

21. - L. 361: Specify which slope is discussed here (it is clear from the figure, but hard to understand from the text).

Response:. We added the description text here to specify the meaning of slope as

**TCD**
"The sum of measured LE and H was significantly less than Rn, with the slope of LE+H versus Rn equal to 0.59 (Fig. 9a)."

22. The language would benefit from editing by a native speaker. Also, references should be checked carefully; references seem to be missing from the reference list (Fisher et al. 2014) or need to be specified (Yu et al. 2016a and 2016b in the reference list, but the text refers to Yu et al. 2016).

Response: We carefully checked the references and made it consistent between the text and reference list. References (Fisher et al. 2014) were added to the reference list. As the changes in the introduction part, Yu et al. (2016a) were no longer there. Accordingly, the newly added references were inserted both in the context and the reference list. The English grammar and fluency have been re-checked carefully throughout the entire manuscript.

---

## Author Comment (AC2) · 31 Jul 2020

We thank the reviewer very much for the constructive comments. We made the point by point response to the comments. The reviewers' comments are in black fonts indexed by numbers. Our response is in blue fonts and red fonts are the updates in the manuscript.

General Comment: "How vadose zone mass and energy transfer physics affects the ecohydrological dynamics of a Tibetan meadow?" by Yu et al. addresses the importance of including freezing and thawing processes and coupling water and energy

balance equations of vadose zone models to simulate ecohydrological dynamics. The authors use data from a high-elevation meadow of the Tibetan plateau to evaluate the outcomes of their different simulations. From their results, the authors claim that adding model complexity improves the model estimates of soil temperature and moisture but may not be necessary to model vegetation dynamics in these regions.

The manuscript is generally well written and understandable. While the presentation and interpretation of the results makes sense, it sometimes lacks clarifications and could benefit from a deeper analysis. I think the model, even fully coupled, does a poor job at reproducing ice content and thermal conditions of deeper layers. As pointed by the authors, some data collected to evaluate the different simulations seem wrong, while others are simply missing, which means it can be barely used to validate any of the simulations. However, the conclusions of this manuscript remain valid and of considerable interest and is valuable to anyone wondering about the importance of including dynamic freezing and thawing processes in their models. Overall, this manuscript has the potential to be an interesting and useful contribution to The Cryosphere, but some critical issues, mostly in the methodology section, need to be addressed before considering publication.

I also think this manuscript does not reach its full potential. It could use various level of complexity in their approach to include freezing and thawing processes. For example, the CPLD simulation could be divided into different simulations: one that only considers latent heat, one that only considers the effect of ice content on the hydraulic conductivity, etc. This would show modelers which component of the freezing/thawing processes is important to consider, and which one is potentially not. However, I acknowledge that this suggestion would involve substantial additional work, so I understand if the authors would rather not make this change.

Response: Many thanks for your constructive comments. We made modifications of the methodology section, in which the clear writing flow was followed. The governing equations and main constitutive equations for different models were clearly presented. The differences among three models regarding soil physical processes were illustrated

in Table 1 and Sect. 3.5. The constitutive equations of unfrozen water content, temperature dependence of liquid flow, the ice effect on hydraulic conductivity, and water vapor density was presented in supplement material (Section S1). We added the geographical location of the Maqu soil moisture and temperature monitoring network and the central experimental site (Figure 1, thus the original figure number was changed). In addition, the description of data gaps was briefly presented. We appreciate your suggested simulations on investigating which component of the freezing/thawing processes is important. This question indeed can be investigated in our future work. We would rather not making this change right now though, and want to present the current results step by step to avoid overwhelming information in one manuscript.

1. Introduction: Well written and introduce well the topic and the problematic. However, I think the authors could add a few lines about the efforts done so far to model freezing and thawing processes in coupled water & energy models for cold regions. There are numerous subsurface and surface models already doing that. They do not all necessarily simulate ecological dynamics, but some studies have provided useful information about the impact of neglecting freezing dynamics.

Response: Thanks a lot for your suggestions. We added in the first part of Introduction some studies that evaluate the modeling of freezing and thawing processes coupled water and energy and pointed out that the novelty here is that also ecohydrological dynamics are jointly evaluated.

Changes in the manuscript: "Concurrently, researchers developed dedicated models, e.g, SHAW (Flerchinger and Saxton, 1989), HYDRUS (Hansson et al., 2004), MarsFlo (Painter, 2011), and STEMMUS-FT (Yu et al., 2018), considering the soil water and heat coupling physics for frozen soils. Promising simulation results have been reported for the soil hydrothermal regimes. While these efforts mainly focus on understanding the surface and subsurface soil water and heat transfer process (Yu et al., 2018) and stress the role of physical representation of freezing/thawing process (Boone et al., 2000;Wang et al., 2017b;Zheng et al., 2017), they seldomly take into account the

interaction with vegetation and carbon dynamics."

2. L56: Unclear what is meant by "changes of frozen ground". Maybe the authors meant something like "variations in seasonally frozen ground thickness"?

Response: Yes, here it is meant to be "variations in seasonally frozen ground thickness". Corrected in the manuscript.

3. L58: What are those divergences? Please provide a few examples.

Response: We added some examples here.

Changes in the manuscript: "However, there are divergences with regard to the expected ecosystem changes across the Tibetan Plateau (Cheng and Wu, 2007;Qin et al., 2016;Wang et al., 2018;Zhao et al., 2010). In response to climate warming, the degradation of frozen ground can positively affect the vegetation growth in Tibetan Plateau mountainous region (Qin et al., 2016), but it can also lead to degradation of grasslands (Cheng and Wu, 2007), depending on soil hydrothermal regimes and climate conditions (Qin et al., 2016;Wang et al., 2016)."

4. L70: I am not sure "complexity" is the best word here. According to the authors, they are testing their models with or without freezing dynamics, and with or without water and energy coupling. As I understand it, there are no different "complexities" in the way frozen soil is represented or in the way the coupling is achieved.

Response: Here we intended to say that the vadose zone water and heat transfer physics is represented in an increasingly complex way (from the T&C without the freezing dynamics, with freezing dynamics but uncoupled water and heat transfer, to the coupling of T&C with STEMMUS to account for fully coupled processes). For unCPLD model (T&C), soil water and heat are independently simulated, ice effect is not considered. For the unCPLD-FT model (T&C-FT), the ice effect on hydraulic conductivity is considered via an impedance factor; thermal conductivity and capacity are affected by the ice content; latent heat change during the freezing/thawing period is

taken into account but heat does not exchange with liquid water. During the non-frozen period, soil water and heat are still independently transferred. For the CPLD model (T&C-STEMMUS), the ice effect on the hydrothermal properties is considered; latent heat change from the freezing/thawing process is also taken into account. Vapor flow, which links the soil water and heat flow, is simulated. The thermal effect on water flow is explicitly considered. Thus, soil water and heat transfer processes are tightly coupled not only during the frozen period but also during the non-frozen period for the CPLD model. We better explained these aspects in Sect. 3.5.

Changes in the manuscript: "To investigate the role of increasing complexity of vadose zone physics in ecosystem functioning, three numerical experiments were designed on the basis of the aforementioned modeling framework (Table 1). First experiment, the T&C original model was run as stand-alone, termed as unCPLD simulation. For the unCPLD model, soil water and heat transfer is independent with no explicit consideration of soil ice effect. The second experiment, the updated T&C model with explicit consideration of freezing/thawing process was run as it can estimate the dynamics of soil ice content and the related effect on water and heat transfer (e.g., blocking effect on water flow, heat release/gain due to phase change) but otherwise being exactly equal to T&C original model. This second simulation is named the unCPLD-FT simulation, where the term unCPLD generally refers to the fact that T&C model and STEMMUS model are not yet coupled. The third experiment, STEMMUS model was coupled with T&C model to enable not only frozen soil physics but also additional processes and most importantly the tight coupling of water and heat effects. This simulation is named CPLD simulation. In this third scenario, vapor flow, which links the soil water and heat flow, is explicitly considered. In addition to the ice blocking effect as presented in unCPLD-FT, the thermal effect on water flow is also expressed with the temperature dependence of hydraulic conductivity and matric potential. Furthermore, not only the latent heat due to phase change, but also the convective heat due to liquid/vapor flow is simulated. "

5. Section 2.1: It is unclear until the reader reaches the results section if the experimental site is underlain by permafrost or not. Please add somewhere in this section that the site only has seasonally frozen ground.

Response: We added this information in the text in Sect. 2.1 as "Seasonally frozen ground is characteristic of this site, with the maximum freezing depth approaching around 0.8 m under current climate conditions."

6. L 93: It is written here that the SMST profiles are measuring temperature and soil moisture at a depth of 80 cm. However, figures from the results section are never showing data at 80 cm, only at 60 cm (which is not listed in line 93). Please fix or clarify.

Response: For Maqu soil moisture and soil temperature (SMST) monitoring network, the SMST profiles are generally measured at depths of 5 cm, 10 cm, 20 cm, 40 cm, and 80 cm. In addition, we have installed some additional 5 TM ECH2O probes to enrich the SMST profile information at specific points. Here, the additional SMST profiles, installed at 2.5 cm, 5 cm, 10 cm, 20 cm, 40 cm, 60 cm, and 100cm, were employed for validating the model simulations. We clarified this point in Sect. 2.1.

Changes in the manuscript: "A few dedicated SMST profiles, with sensors installed at depths of 2.5 cm, 5 cm, 10 cm, 20 cm, 40 cm, 60 cm, and 100 cm, were used for validating the model simulations."

7. Figure 1b: If the figure is indeed showing both freezing and thawing fronts, I would recommend using different colors. It is unclear which zone is thawed and which one is frozen (mostly in 2017-2018). I also think the y-axis should not reach 100 cm. Based on the text, there is no sensor deeper than 80 cm. If I understood correctly, the graph is plotting information the authors do not have. Please fix or clarify in the text. Furthermore, is the data here interpolated from the 5 sensors described in line 93?

Response: Different colors are now presented in Figure 2b, with the freezing fronts

color blue and thawing fronts color red. In the updated Figure 2b, soil temperature measurements at depths of 2.5 cm, 5 cm, 10 cm, 20 cm, 60 cm, and 100 cm were used to generate the freezing/thawing front propagation dynamics. Soil temperature measurements at 40cm are not used as its large data gaps during 2017-2018 wintertime (see Table R.1).

Table R.1. The main data gaps for soil moisture/soil temperature (SMST) measurements

| SMST | Data gaps |
|---|---|
| 2.5 cm | 3/25/2016 - 06/08/2016, 3/29/2017 - 27/07/2017 |
| 5 cm | 3/25/2016 - 06/08/2016, 3/29/2017 - 27/07/2017 |
| 10 cm | 3/25/2016 - 06/08/2016, 3/29/2017 - 27/07/2017 |
| 20 cm | 3/25/2016 - 06/08/2016, 3/29/2017 - 27/07/2017 |
| 40 cm | 3/25/2016 - 06/08/2016, 3/29/2017 - 12/08/2018 |
| 60 cm | 3/25/2016 - 06/08/2016, 3/29/2017 - 27/07/2017 |
| 100 cm | 3/25/2016 - 06/08/2016, 3/29/2017 - 27/07/2017 |

Changes in the manuscript: "A few dedicated SMST profiles, with sensors installed at depths of 2.5 cm, 5 cm, 10 cm, 20 cm, 40 cm, 60 cm, and 100 cm, were used for validating the model simulations. Note that there are data gaps (25th Mar – 8th June, 2016; 29th Mar – 27th July, 2017, extended to 12th Aug, 2018 for 40 cm) due to the malfunction of instruments and the difficulty to maintain the network under harsh environment."

8. L 124: Please provide more information about the missing data. When and where? This should also be shown in Figure 1b. For example, the authors can use a greyed area to show where data are missing. This is crucial considering the authors are using this data or comparison purposes in Figure 6.

Response: The main data gaps for SMST measurements are summarized as Table

R.1. For Figure 1b, the propagation of freezing/thawing front was obtained by interpolated the SMST profile measurements (2.5 cm, 5 cm, 10 cm, 20 cm, 60 cm, and 100 cm). Here, soil temperature measurements at a depth of 40 cm were omitted as its long data gaps during the second winter period (2017-2018).

Changes in the manuscript: "Note that there are data gaps (25th Mar – 8th June, 2016; 29th Mar – 27th July, 2017, extended to 12th Aug, 2018 for 40 cm) due to the malfunction of instruments and the difficulty to maintain the network under harsh environment."

9. Equation 1: Check units. Are the units of S really s-1 or rather kg m-3 s-1?

Response: The unit of S, the sink term, is kg m-3 s-1 with the consideration of water density. We corrected the unit of S as 'kg m-3 s-1'.

10. Equation 2: Undefined variables: dz, rH, rL, nc

Response: Equation 2: Note that to avoid confusion, the terms not used in this simulation case was deleted from Eq. 2. We added the relevant description.

Changes in the manuscript: $d_{z,i}\frac{d\theta_i}{dt} = q_{i-1} - q_i - T_v r_{v,i} - E_s - E_{bare}$ (2)
where $d_{z,i}$ (m) is the thickness of layer i; $q_i$ $(ms^{-1})$ is the vertical outflow from a layer i; $T_v$ $(ms^{-1})$ is the transpiration fluxes from the vegetation; $r_{v,i}$ is the fraction of root biomass contained in soil layer i; $E_{bare}$ $(ms^{-1})$, evaporation from the bare soil; $E_s$ $(ms^{-1})$, evaporation from soil under the canopy.

11. Equation 4: Csoil is presented as the specific heat capacity of the bulk soil, but I think it is the volumetric heat capacity. The units do not match otherwise. The same issue arises in equations 6, 7 and 8. Equation 8 is from Hansson et al. (2004), where Csoil is defined as the volumetric heat capacity.

Response: Many thanks for pointing out this. Equation 4: There was an inconsistency between the left and right side of Equations 4, 6, 7, and 8 (now Eqs. 3, 4, 5, 6). We multiplied Csoil with the soil density $\rho_{soil}$ to make it consistent. Capp was clarified as

the apparent volumetric heat capacity here as Equation 8 (now Eq. 5).

12. Equation 8: Please define which phase is represented by $d\theta$. It should be liquid water.

Response: Thanks a lot for your comment. We added some text as "differential (specific) water capacity $d\theta/d\varphi$ at a given liquid water content $\theta$".

13. Section 2.4.4: The authors describe in detail the equations of the two models but do not explain the critical components of the added freezing/thawing processes, except for latent heat. They do provide references but, considering that this is supposedly an important aspect of this manuscript, they should at least describe in more details the different equations used. For example, the calculation of the hydraulic conductivity for a frozen medium is quite important. The authors refer here to Hansson et al. (2004), which uses an impedance factor. This method is widely used, but the impedance factor is an arbitrary number that is likely to change based on the type of soil. The authors never write which impedance factor is used. I am also concerned that the authors are using the method from Dall'Amico et al. (2011) for the soil freezing characteristic curve and the method from Hansson et al. (2004) for the apparent heat capacity. These two papers both use a form of the Clausius-Clapeyron equation with the van Genuchten model, but with different approaches. I could be wrong but, depending on how they have been used, these two methods may not be compatible with each other. More information is required here to make sure the freezing model used by the authors is valid.

Response: We added the description of the freezing/thawing process, i.e., how unfrozen water content, hydraulic conductivity is calculated, the assignment of impedance factor, temperature dependency of hydraulic conductivity/matric potential, vapor density parameterization. As this information does not represent the main physical equation but rather parameterizations, we did not list them in the main text but in the supplemental materials (Supplement S1).

The apparent heat capacity is calculated as $C_{app} = \rho_{soil}C_{soil} - \rho_{ice}L_f\frac{\partial\theta_{ice}}{\partial T}$ to express both the heat conduction and latent heat (phase change) terms. On the basis of the assumption of zero ice gauge pressure and osmotic pressure, Clausius-Clapeyron equation is utilized to convert $-\rho_{ice}L_f\frac{\partial\theta_{ice}}{\partial T}$ into $\rho_{ice}\frac{(L_f^2)}{gT}\frac{d\theta}{d\varphi}$. This is a general procedure to derive the apparent heat capacity and it is independent of which form of the soil-freezing characteristic function is used. The assumption and the Clausius-Clapeyron equation used in Dall'Amico (2011) is the same as Hansson et al. (2004) (i.e., $\frac{dP}{dT} = \frac{L_f}{V_w T}$), there is no difference between these two papers in calculating the apparent heat capacity. The beauty of the equation (19) – (21) in Dall'Amico et al., (2011) - the equations used to estimate the additional freezing pressure due to the unfrozen water pressure - is that they are fully energy conservative (contrary to other formulations) and are also independent of the specific hydraulic parameterization of the soil (even though later on van Genuchten method is used in Dall'Amico et al. 2011). As a matter of fact, in T&C-FT, three different soil hydraulic parameterizations can be used: (i) van Genuchten, (ii) Saxton-Rawls and (iii) Clapp and Hornberger. For all of these the liquid water pressure is derived with equation (21), but then the actual liquid and frozen water content depend on the specific soil hydraulic parameterization chosen. Before implementing the Dall'Amico et al., (2011) formulation, we had a lot of problems related to energy conservation in the freezing/melting phases, problems that were not apparent for a week-ten days of simulations but developed over time, so we checked carefully that energy is indeed properly conserved in the long-term in T&C-FT.

Changes in the manuscript: "The soil freezing characteristic curve providing the liquid water potential in a frozen soil is computed following the energy conservative solution proposed by Dall'Amico et al. (2011) and it can be combined with various soil hydraulic parameterizations including van Genuchten and Saxton and Rawls to compute the maximum liquid water content at a given temperature and consequently ice and liquid content profiles at any time step (Fuchs et al., 1978;Yu et al., 2018)."

14. Section 2.6: I think this section lacks some important clarifications. First, I think the

differences between the models are more complicated than coupled/uncoupled. The vadose zone equations of T&C model are not coupled, because the water and heat equations are independent from each other. On the other end, the heat equation of STEMMUS needs to be coupled to the mass transfer equation because of the consideration of different processes or constituents such as heat advection. However, when adding the freezing/thawing processes, the heat and water equations of T&C becomes somehow coupled (at least one-way) due to the temperature dependency of the hydraulic conductivity. It is unclear if the authors used the names unCPLD, unCPLD-FT and CPLD for their models to characterize the way the heat and water equations are solved or to characterize how the two components (T&C and STEMMUS) are used. If it is the former, I suggest that they authors rename their simulations as T&C, T&C-FT and T&C-STEMMUS. In any case, the coupling characteristics of the different simulations should be further explained. Secondly, it is unclear which processes/parameters are considered in each simulation. For example, the authors state in the discussion that "unCPLD-FT simulation accounting for soil-freezing in a simplified way in comparison to STEMMUS (e.g., the CPLD simulation)" (line 418-419). However, to my knowledge, the difference in the way STEMMUS accounts for soil-freezing processes is never explained. A paragraph describing the different processes that each simulation is accounting for is necessary in this section. This can also take the form of a table.

Response: Thanks a lot for the comment. Yes, the "coupling" in the label was referring to the coupling among models rather than in the processes. In order to present the difference among the model versions and clarify the words 'unCPLD, CPLD', we added Table 1 and relevant descriptions in the Sect. 3.5 to explain the difference and the processes considered in each simulation case.

15. Figure 3: The grey line is hard to distinguish. I suggest making it slightly darker or thicker.

Response: We made the grey line darker and thicker in Figure 4.

16. Figures 5 and 6: It looks like deeper soil moisture and temperature is not well reproduced by any of the models, even though it is not discussed in the text. This has the consequence of poorly representing ice content (Figure 7), mostly in 2017-2018. It is understandable considering the model has not been calibrated and the goal of this manuscript, which focuses more on the growing season, is not necessarily to validate the models. However, I think this poor fit with field measurement should be discussed in the text. There are many reasons that could explain this, such as the presence of heterogeneity in the soil or of freezing-point depression due to increased salinity.

Response: In Sect. 4.2, We now discussed the reasons for the discrepancies between the model simulated and observed soil moisture and temperature at deeper soil layers.

Changes in the manuscript: "It should be noted that for the deeper soil layers (e.g., 60cm in Figure 7), all models tended to simulate the early start of freezing soil temperatures and considerably underestimated the soil temperature during the frozen period. This can be due to the uncertainties in soil organic layer parameters, the not fully captured snow cover effect (Gouttevin et al., 2012), a potentially pronounced heterogeneity in soil hydrothermal properties, or the potential role of solutes on the freezing-point depression (the presence of solute lowers the freezing soil temperature) (Painter and Karra, 2014). These mismatches in deep soil temperature degraded the model performance in simulating the dynamics of liquid water (Figure 6) and ice content (Figure 8) during the frozen period."

17. L 303: I think they authors meant "unCPLD-FT" instead of "unCPLD".

Response: We initially used "unCPLD" model simulations to refer to both the "unCPLD" and "unCPLD-FT" model simulations, as Fig. 8 also presents the water flow simulations of unCPLD model. To avoid confusion, we rephrased this sentence as "The time-series of soil ice content and water flux from unCPLD, unCPLD-FT and CPLD model simulations for soil layers below 2 cm are presented in Figure 8."

18. L 307-309: I do not agree that the CPLD model shows a good match with field

[Figure]

measurements of ice content, at least not as currently showed in winter 2017-2018.

Response: Yes. Both the unCPLD and CPLD model cannot well reproduce the soil ice content based on current simulations. Here, we would like to stress the difference between unCPLD-FT and CPLD model. CPLD model presents a shallower freezing ice depth compared to unCPLD-FT model. We attributed this to the physical difference between unCPLD-FT and CPLD model, i.e., the constraints by the interdependence of liquid, ice, vapor, air components in the soil pores are considered by CPLD model.

Changes in the manuscript: "It is to note that compared to unCPLD-FT model, CPLD model presented a relatively lower presence of soil ice content, while its temporal dynamics was closer to the observed freezing/thawing front propagation. The difference between the two simulations can be attributed to the constraints imposed by the interdependence of liquid, ice and vapor in the soil pores that is considered only in CPLD model. "

19. Figure 8: Please define acronyms in the caption (e.g., "(a) Gross Primary Production" instead of "(a) GPP")-friendly version

Response: We describe acronyms in the caption of Figure 9.

Changes in the manuscript: "Figure 9. Comparison of observations from Eddy Covariance (OBS) or MODIS remote sensing and simulated (a) Gross Primary Production (GPP), (b) Leaf Area Index (LAI), (c) Net Ecosystem Exchange (NEE), and (d) Ecosystem respiration (Reco) using unCPLD, unCPLD-FT, and CPLD model. MODIS refers to the data from MODIS-GPP and MODIS-LAI products."

20. Figure 9: The authors use R2 here and R in Figure 8. I suggest them to choose one and be consistent.

Response: Yes, sorry for the confusion. We use the determination coefficient R2 and keep it consistent in Figure 5 and Figure 10.

21. Figure 10: Please define acronyms in the figure caption.

[Figure]

Response: We added the descriptions of the acronyms as "$T_v$, transpiration; $E_s$, surface evaporation; $E_{IN}$ and $E_{SN}$, evaporation from intercepted canopy water and snow cover; $\Delta Vs$, changes in soil water storage; $L_K$ , deep leakage water.".

22. L 381-382: Confusing sentence. Should we compare to unCPLD or unCPLD-FT?

Response: Here we compare unCPLD model simulations with unCPLD-FT model simulation to highlight the role of ice content and latent heat associated with phase change only. This has been clarified as"Less amount of water was consumed by ET from unCPLD-FT simulations than that from unCPLD."

23. L 383-384: What could explain cooler late winter temperatures in unCPLD-FT? Latent heat slowing down the thawing? Lower bulk heat capacity of frozen soil? Please provide hypotheses.

Response: As for unCPLD-FT model, the effect of ice content and latent heat due to phase change were taken into account, while these two effects were absent by unC-PLD model. During the late winter (mainly thawing periods), the thawing process was retarded by the heat absorption due to phase change. Secondly, the ice induced soil heat capacity also damped the magnitude of temperature variations, make the thawing process of soil temperature more difficult and slower. Generally, as discussed earlier there is a quite evident cryoscution in the unCPLD-FT simulation that generates a considerable amount of ice content in the soil that takes time to be melt and reduce the average temperature. Thus there are cooler late winter temperatures in the unCPLD-FT model than in the unCPLD simulations.

Changes in the manuscript: "The cooler late winter temperatures from unCPLD-FT simulations can be attributed to the retardation of the thawing process due to the phase change-induced heat absorption and the soil ice-induced modification of bulk heat capacity during the freezing-thawing transition period, which damped the magnitude of temperature variations and delayed the thawing process."

24. L 396-400: This analysis could be improved. The coupling is not the only difference between CPLD and unCPLD-FT. STEMMUS is simulating some subsurface processes that T&C does not (e.g., heat advection, air flow, vapor flow). I recommend providing a more detailed analysis then simply justifying the differences by the coupling. Also, how is ice content and hydraulic conductivity being simulated differently in CPLD than in unCPLD-FT?

Response: We agree with your points that the coupling is not the only difference between CPLD and unCPLD-FT model. These contexts (L396-400) were used to explain why the water storage amount in the vadose zone is increased while the bottom leakage decreased for CPLD model (Figure 11). It lies in the difference of the considered subsurface processes between CPLD and unCPLD models (Sect. 3.5, Table 1). We further described the hydraulic conductivity (and its temperature dependency), vapor density in supplement S1.
Hydraulic conductivity in CPLD (T&C-STEMMUS) model is dependent on temperature in two ways. 1. The impedance effect of soil ice content on the hydraulic conductivity , which is dependent on soil temperature, (i.e., reducing the saturated hydraulic conductivity via an empirical impedance factor). 2. The water viscosity effect on hydraulic conductivity. As the temperature decreases, water movement slows down.

Changes in the manuscript: "We attribute this to the way ice content is simulated in the CPLD simulation, and also to the temperature dependence of soil hydraulic conductivity (see Table 1 and Supplement S1). Specifically, the high accumulation of ice content in the unCPLD-FT simulations indicates a relatively stronger cryosuction effect than in CPLD simulations. This cryosuction effect is mitigated in the fully coupled model because of water vapor transfer and thermal gradients, even though different solutions in the parameterization of bulk soil thermal conductivity and volumetric soil heat capacity could also be responsible for the difference. Overall, taking into account the fully coupled water and heat physics modify the temporal dynamics of ice formation and thawing in the soil and activates temperature effects on water flow (i.e., low soil

temperature will slow down water movement)."

25. L 407-410: There are two requirements to experience heat advection: water flow and difference in temperature. While the former is shown in Figure 7, there is no evidence shown for the latter. It would be interesting if the authors could provide some evidence (can be with numbers or words) that heat advection (or convective heat) is mostly relevant during the frozen period.

Response: Here, we highlight the difference between unCPLD and CPLD models on heat advection effects. As shown in Figure 8c and d/e, the difference in water flow can be several orders of magnitude. We added some references here to corroborate this point.

Changes in the manuscript: "The liquid water flux-induced convective heat flux is mostly relevant during the frozen period (Boike et al., 2008;Kane et al., 2001;Yu et al., 2020). As it has been observed, a certain amount of liquid water/vapor flux moving toward the freezing front and this effect is different between unCPLD-FT and CPLD while absent in unCPLD (Figure 8). For the unfrozen period, instead, the total mass fluxes were comparable between the two unCPLD and CPLD simulations. For the temperature gradient, there is not much difference between unCPLD and CPLD simulations during both the growing season and frozen period."

26. L 413: I think Figure 8 should be referred here instead of Figure 9.

Response: Yes sorry. We corrected it in the manuscript (Line 473). Thanks a lot.

[Figure]

References

[revised manuscript text omitted]

**Supplement:**

**Supplement**

In this supplement, we first presented the constitutive equations regarding unfrozen water content, the ice effect on hydraulic conductivity, the temperature dependence of water flow, water vapor density in Section S1. Then Section S2 presented the surface fluxes simulations for the frozen/unfrozen periods. Tables were

5    listed in Section S3.

**S1 Constitutive equations**

**S1.1 Unfrozen water content**

In both T&C and STEMMUS, the soil freezing characteristic curve (SFCC) method was employed to estimate unfrozen water content, in combination with the van Genuchten soil water retention curve

10   (SWRC) model (Van Genuchten, 1980) and Clapeyron equation. The SWRC is expressed as

$$\theta_{tot}(h) = \begin{cases} \theta_r + \frac{\theta_s - \theta_r}{[1+|\alpha h|^n]^m}, & h < 0 \\ \theta_s, & h \geq 0 \end{cases}, \tag{S1}$$

where $\theta_{tot}$, $\theta_s$, and $\theta_r$ are the total water content, saturated water content and the residual water content, respectively; $h$ (m) is the pre-freezing soil water potential; $\alpha$ is related to the inverse air-entry pressure; $m$ is the empirical parameter. The parameter $m$ is a measure of the pore-size distribution and can be expressed as $m = 1$-$1/n$, which in turn can be determined by fitting van Genuchten's analytical model (Van Genuchten,

15   1980).

The unfrozen water content was estimated by employing SFCC (Dall'Amico, 2010; Dall'Amico et al. 2011)

$$\theta_L(h,T) = \theta_r + \frac{\theta_s - \theta_r}{[1+|\alpha(h+h_{Frz})|^n]^m}, \tag{S2}$$

where $\theta_L$ is the liquid water content, $L_f$ (J kg$^{-1}$) is the latent heat of fusion, $g$ (m s$^{-2}$) is the gravity acceleration, $T_0$ (273.15 °C) is the absolute temperature. $h$, $\alpha$, $n$, and $m$ are the same as in S1. $h_{Frz}$ (m) is the soil freezing potential.

$$h_{Frz} = \frac{L_f}{gT_0}(T - T_0) \cdot H(T - T_{CRIT}), \tag{S3}$$

20   where $T$ (°C) is the soil temperature. $H$ is the Heaviside function, whose value is zero for negative argument and one for positive argument, $T_{CRIT}$ (°C) is the soil freezing temperature.

$$T_{CRIT} = T_0 + \frac{ghT_0}{L_f},$$
(S4)

**S1.2 Hydraulic conductivity**

In both T&C and STEMMUS, the unsaturated hydraulic conductivity (Van Genuchten, 1980, Mualem 1976) is expressed as

$$K_{Lh} = K_s K_r = K_s S_e^l [1 - (1 - S_e^{1/m})^m]^2,$$
(S5)

$$S_e = \frac{\theta - \theta_r}{\theta_s - \theta_r},$$
(S6)

$$m = 1 - 1/n,$$
(S7)

where $K_{Lh}$, $K_s$ and $K_r$ (m s$^{-1}$) are hydraulic conductivity, saturated hydraulic conductivity and relative hydraulic conductivity, respectively. $S_e$ is the effective saturation. $l$, $n$, and $m$ are the van Genuchten fitting parameters.

The blocking effect of ice presence is estimated by the impedance factor,

$$K_{fLh} = 10^{-EQ} K_{Lh},$$
(S8)

$$Q = (\rho_i \theta_{ice}/\rho_L \theta_L),$$
(S9)

where $K_{fLh}$ (m s$^{-1}$) is the hydraulic conductivity in frozen soils; $K_{Lh}$ (m s$^{-1}$) is the hydraulic conductivity in unfrozen soils at the same negative pressure or liquid moisture content; $\theta_{ice}$ is soil ice content; $Q$ is the mass ratio of ice to total water, and $E=7$ is the empirical constant that accounts for the reduction in permeability due to the formation of ice (Hansson et al., 2004).

**S1.3 Temperature dependence of matric potential and hydraulic conductivity**

Soil matric potential and hydraulic conductivity are dependent on temperature in STEMMUS (Zeng and Su, 2013), which affects soil water surface tension and viscous flow effects. The temperature dependence of matric potential can be expressed as

$$h_{Cor\_T} = he^{-C_\psi(T-T_r)}$$
(S10)

where, $h_{Cor\_T}$ is the soil matric potential considering temperature effect; $C_\psi$ is the temperature coefficient, assumed to be constant as 0.0068 °C$^{-1}$ (Milly, 1984); $T_r$ is the reference temperature (20 °C).

Hydraulic conductivity, taken into account the temperature effect, can be written as

$$K(\theta, T) = K_s K_r(\theta) K_T(T) \tag{S11}$$

40    where $K_r(\theta)$ is the relative hydraulic conductivity, $K_T(T)$ is the temperature coefficient of hydraulic conductivity, expressed as

$$K_T(T) = \frac{\mu_w(T_r)}{\mu_w(T)} \tag{S12}$$

where $\mu_w$ is the viscosity of water. The dynamic viscosity of water can be written as

$$\mu_w = \mu_{w0} \exp\left[\frac{\mu_1}{R(T + 133.3)}\right] \tag{S13}$$

where $\mu_{w0}$ is the water viscosity at the reference temperature, $\mu_1$=4.7428 (kJ mol⁻¹), $R$ =8.314472 (J mol⁻¹ ºC⁻¹), T is the temperature in ºC.

45    **S1.4 Water vapor density**

Vapor flow, driven by the gradient of water vapor density, links the water flow and heat flow in STEMMUS. The water vapor density, according to Kelvin's law, is expressed as a function of both temperature and matric potential (Philip and Vries, 1957)

$$\rho_V = \rho_{sV} H_r, \quad H_r = \exp\left(\frac{hg}{R_V T}\right), \tag{S14}$$

where $\rho_{sV}$ is the density of saturated water vapor ($\exp\left(31.3716 - \frac{6014.79}{T} - 7.92495 \times 10^{-3}T\right)\frac{10^{-3}}{T}$); $H_r$

50    is the relative humidity; $R_V$ (461.5 J kg⁻¹ K⁻¹) is the specific gas constant for vapor; $g$ is the gravitation acceleration; $T$ is the temperature in K.

55

**S2 Surface Fluxes Simulations**

The difference between model simulated and the eddy covariance measured 5-day moving average dynamics of surface energy fluxes was shown in Figure S1. Three models generated similar simulation patterns against the observations. The simulation of $Rn$ is characterized as the overall overestimation, with isolated underestimation episodes, which occurred mainly in the frozen periods. Compared with unCPLD and unCPLD-FT model simulations, CPLD model presented a larger overestimation of $Rn$. Such overestimation tended to enlarge during the non-growing periods. There is a negligible difference of $Rn$ between unCPLD and unCPLD-FT model simulations.

For H dynamics, model differences presented the seasonal variations with the general overestimation during the growing seasons while the underestimation during the non-growing seasons. During the growing season, the modeling discrepancies appeared larger in 2016 and 2018 than that in 2017. It is difficult to attribute such a difference mostly to the model inaccuracy or mostly to the data inaccuracy. On one hand, the energy balance closure problem rises as the potential source of error and reason of discrepancy. The Eddy covariance observed LE and mostly H fluxes are underestimated when constrained by the surface energy closure during the summer periods (see Table 2). On the other hand, in T&C model, the surface temperature is simplified and 'one single prognostic surface temperature' is computed, i.e. soil surface and vegetation surface temperature have the same value. The difference between the soil surface, vegetation surface and the assumed surface temperature can be a potential cause for such discrepancies in H. In addition, the uncertainties in the precipitation measurements can be an additional potential reason for the simulated differences between 2016 & 2018 and 2017.

CPLD model generated less overestimation of H, compared with unCPLD and unCPLD-FT model. As the main difference among the three models is that CPLD model taking into account the coupling water and heat physics during the unfrozen period. We attributed such difference to the water-heat coupling physics, i.e., the vapor flow effect and thermal effect on liquid flow. During the frozen periods, CPLD model usually produced a larger underestimation than unCPLD and unCPLD-FT models. Slightly better performance was identified during the late winter periods for CPLD model, probably due to the better capture of vegetation dynamics.

There are seasonal fluctuations of model performance regarding LE dynamics, with the general overestimation during the growing season while good fits during the non-growing season. The differences among the three models were minimal except some observable differences during the vegetation onset periods when the difference of vegetation dynamics occurred.

The correlation between observed and model simulated daily average surface energy fluxes for the non-frozen and frozen period was presented as Figure S2 and Figure S3. For the non-frozen period, three models can well simulate the dynamics of $Rn$ except at the low radiation values. The correlation between model simulated and measured LE was weaker than $Rn$. The worst model performance was identified for H

90   simulations. Three models produced a similar correlation to the observed surface fluxes during the non-frozen period.

For the frozen period, the model performance degraded for the surface energy fluxes. There is a considerable underestimation of *Rn* against the measured high *Rn* values. This is probably due to the snow cover dynamics were not well captured and the uncertainties in precipitation measurements. Other than these periods, CPLD
95   model produced overestimation of *Rn*, which results in a worse correlation than that from unCPLD and unCPLD-FT models. There is no significant difference among the three models in LE simulations. The correlation between model simulated and observed H dynamics appeared the same for the three models, while CPLD model produced the underestimation of H compared with unCPLD and unCPLD-FT simulations.

[Figure]

**Figure S1. Difference between observed and simulated 5-day moving average dynamics of net radiation (Rn), latent heat flux (LE), and sensible heat flux (H) using the original (uncoupled) T&C (unCPLD), T&C with consideration of FT process (unCPLD-FT) and coupled T&C and STEMMUS (CPLD) model. The frozen period, identified from Figure 1b, was highlighted by the blue shadow.**

 **Non-frozen period**

[Figure]

**Figure S2. Scatter plots of observed and model simulated daily average surface fluxes (net radiation: Rn, latent heat: LE and sensible heat flux: H) using the original (uncoupled) T&C (unCPLD), T&C with consideration of FT process (unCPLD-FT) and coupled T&C and STEMMUS (CPLD) model during the non-frozen period, with the color indicating the occurrence frequency of surface flux values.**

110    **Frozen period**

[Figure]

**Figure S3. Scatter plots of observed and model simulated daily average surface fluxes (net radiation: Rn, latent heat: LE and sensible heat flux: H) using the original (uncoupled) T&C (unCPLD), T&C with consideration of FT process (unCPLD-FT) and coupled T&C and STEMMUS (CPLD) model during the frozen period, with the**
115    **color indicating the occurrence frequency of surface flux values.**

**S3 Tables**

**Table S1. The average values of soil texture and hydraulic properties at different depths used in all simulations.**

| Soil depth (cm) | Clay (%) | Sand (%) | $K_s$ ($10^{-6}$ m s$^{-1}$) | $\theta_s$ (cm$^3$ cm$^{-3}$) | VG model $\theta_r$ (cm$^3$ cm$^{-3}$) | $\alpha$ (m$^{-1}$) | $n$ |
|---|---|---|---|---|---|---|---|
| 5-10 | 10.00 | 27.00 | 1.05 | 0.55 | 0.050 | 0.015 | 1.35 |
| 10-40 | 8.00 | 28.00 | 1.94 | 0.55 | 0.050 | 0.008 | 1.45 |
| 40-80 | 8.00 | 47.00 | 5.61 | 0.50 | 0.052 | 0.008 | 1.50 |

Note: VG, Van Genuchten (Van Genuchten, 1980)

120

**Table S2. The main vegetation parameters for the Tibetan meadow ecosystem used in all simulations**

| Parameter | Symbol | Unit | Value |
|---|---|---|---|
| Root depth that contains 95% of fine root biomass | $Z_{R,95}$ | m | 0.3 |
| Water use efficiency parameter, which connects the stomatal aperture and net assimilation | $a_1$ | - | 5 |
| Specific leaf area | $S_{LAI}$ | m$^2$ LAI g C$^{-1}$ | 0.0225 |
| Maximum rubisco capacity | $Vc_{max}$ | - | 60 |
| Temperature for leaf onset | $T_{lo}$ | ºC | 0.2 |
| Daylight threshold for senescence | $L_{day\_cr}$ | h | 11.4 |
| Cold control on leaf shedding | $T_{cold}$ | ºC | 0 |
| Water potential at 2% loss stomatal conductivity | $\psi_{S,00}$ | MPa | -0.8 |
| Water potential at 50% loss stomatal conductivity | $\psi_{S,50}$ | MPa | -2.8 |
| Critical leaf age | $A_{cr}$ | d | 180 |
| Leaf onset water stress | $\beta_R$ | - | 0.99 |

**Table S3. A summary of annual average GPP and NEE of grassland over the Tibetan Plateau, with records of mean annual temperature (MAT) and precipitation (APT)**

| Site | Location | Elevation (m) | Type | MAT (°C) | APT (mm) | GPP (gC m$^{-2}$ yr$^{-1}$) | NEE (gC m$^{-2}$ yr$^{-1}$) | Reference |
|---|---|---|---|---|---|---|---|---|
| Arou | 38°3′N, 100°27′36″E | 3033 | grassland | 0.6 | 464.1 | 818.3 | -198.7 | (Sun et al., 2019) |
| Damxung[a] | 30°28′08.50″ N, 91°03′44.50″E | 4286 | swamp meadow | - | - | 755-935 | - | (Bai et al., 2011) |
| Damxung[a] | 30°28′08.50″ N, 91°03′44.50″E | 4286 | swamp meadow | 1.8 | 475.6 | 835.29 (755.02-901.37) | - | (Niu et al., 2016) |
| Damxung[a] | 30°28′08.50″ N, 91°03′44.50″E | 4286 | swamp meadow | 1.3 | 335 | - | -161.85 | (Niu et al., 2017) |
| Haibei[a] | 38°37′N, 101°18′E | 3250 | shrub meadow | -1 | 566 | 634 (575-681) | -121(-193 ~ -79) | (Kato et al., 2006) |
| Haibei[a] | 37°35′N, 101°20′E | 3250 | alpine wetland meadow | -1.1 | 510.367 | 629.87 (575.7-682.9) | 106.1 (44-173.2) | (Zhao et al., 2010) |
| Lijiang | 27°100′ N, 100°140′ E | 3560 | alpine meadow | 6.1 | 1180 | 600 (522-669) | -161 (-213 ~ -114) | (Wang et al., 2017) |
| Zoige[a] | 33°56′ N, 102°52′ E | 3430 | swamp meadow | - | - | 589.8-672.1 | - | (Tian et al., 2003) |
| Zoige[a] | 33°56′ N, 102°52′ E | 3430 | swamp meadow | 1.1 | 650 | - | -79.7~-47.1 | (Hao et al., 2011) |
| 10 sites | - | 3033-4730 | alpine grassland | - | - | 300-400 | - | (He et al., 2014) |

125    Note: [a] indicates same site with different years

---

## Author Response (AR2)

We thank the editor and referees very much for the time in processing and commenting our manuscript. We made our point-by-point response as below in blue font. The updates in the manuscript are in red font.

**Referee #1**

*General comments:* The revised version of the manuscript has improved substantially, and I would like to thank the authors for their efforts to address my comments. The description of the three model versions is much more understandable and it is now made very clear what the differences between the three model versions are. I also appreciate the additional figures in the supplementary material that help to evaluate the three model versions separately.

I have two major comments left (one regarding the manuscript itself, and one regarding the measurement data used), and have a number of smaller remarks that may help to improve the manuscript. Once these are taken into account, I consider the manuscript acceptable for publication.

Thanks a lot for your dedicated comments, which are very helpful in improving this manuscript. Please find our response as below.

Major remarks:

1. In the introduction, relevant studies are mentioned as examples of modelling studies on relevant ecosystems, but merely as a list of examples without reference to their outcomes. They are listed to have shortcomings because processes are "independent and not fully coupled" – it would be great to have this argument explained further based on the referenced studies, because the manuscript addresses exactly these shortcomings.

**Response**: We rephrased these sentences and briefly integrated these researches as "By either incorporating a permafrost model into the ecosystem model (Lyu and Zhuang, 2018; Wania et al., 2009; Zhuang et al., 2001) or equipping the soil model with vegetation dynamics and carbon processes (Zhang et al., 2018), the temporal dynamics of soil temperature, permafrost dynamics and vegetation and carbon dynamics can be simultaneously simulated over cold region ecosystems. Moreover, the incorporation of detailed vadose zone and land surface processes (e.g., soil hydrology and snow cover) usually improves the model performance (Lyu and Zhuang, 2018) and facilitates model ability to investigate the ecosystem response to variations in climatic and environmental conditions at various spatial-temporal scales (Zhang et al., 2018). The

importance of non-growing-season processes (e.g., freeze-thaw cycle, snow cover) was highlighted when interpreting the carbon budget observations and can significantly alter the carbon cycling and future projection of cold region ecosystems (Lyu and Zhuang, 2018; Wania et al., 2009; Zhang et al., 2018; Zhuang et al., 2001).".

2. It is not clear to me whether the observed turbulent fluxes in Fig. 4 were determined from the eddy covariance system (L. 115) or from energy balance computations – the manuscript mentions only the eddy covariance system, but there is a striking similarity in the time series of Rn and LE (Fig. 4a and 4c), hinting at the possibility that LE is computed based on Rn instead. If these data are based on eddy covariance, please explain how processing and gap filling was done (similar to the description of the carbon fluxes, section 2.2.1), and whether the measured Rn is used in any way for this.

**Response**: The observed turbulent fluxes in Figure 4 were determined from the eddy covariance system. Radiation is used to define the similar meteorological conditions for the gap filling. The average values of LE under these similar meteorological conditions were used to replace the missing values.

We added the relevant text describing how the processing and gap filling was done.

"Then, considering the covariation of fluxes with meteorological variables and the temporal auto-correlation of fluxes, the marginal distribution sampling algorithm was used as the gap-filling method to replace the missing data (Reichstein et al., 2005). Three cases were identified according to the availability of $R_g$, $T_{air}$, and $VPD$: Case 1, $R_g$, $T_{air}$, and $VPD$ data are available; Case 2, only $R_g$ data are available; Case 3, none of the $R_g$, $T_{air}$, and $VPD$ data are available. A look-up table (LUT) method was used to search for the similar meteorological conditions (i.e., under which $R_g$, $T_{air}$, and $VPD$ do not deviate by more than 50 W m$^{-2}$, 2.5 °C, and 5 hPa, respectively, for case 1) within a certain time window. The average value of *NEE* under these similar meteorological conditions was used to replace the missing gaps. The time window size started from 7 days and extended to 14 days if no similar meteorological conditions were detected. The similar LUT approach was utilized for case 2, the similar meteorological conditions were determined only by $R_g$ within a time window of 7 days. For case 3, the missing value of *NEE* was replaced by the average value of adjacent hours (within 1 hour) at the same day or at the same time of the day, which was derived from the mean diurnal course within 2 days. The aforementioned three steps were repeated with increased window sizes until the missing value could be properly filled. Finally,

*NEE* was separated into *GPP* and $R_{eco}$ by night-time based and day-time based approaches (Lasslop et al., 2010). Land surface energy fluxes (*LE*, *H*) were processed simultaneously using the aforementioned $u_*$ filtering and gap filling methods with REddyProc package."

Minor remarks:

3. - L. 12: "… even more so in cold regions…": It is not clear to me why vadose zone would be more crucial in cold regions than in others – there are many other ecosystem types where the vadose zone can also be attributed a crucial role for ecosystem processes (e.g. in subtropical arid and semi-arid ecosystems).

**Response**: Yes. We agree that the vadose zone can also be attributed a crucial role for ecosystem processes in subtropical arid and semi-arid ecosystems.

Here we intended to compare the role of vadose zone in cold regions under the current situation with that under the future climate warming conditions, and thus highlight the necessity of this study.

"… even more so in cold regions with seasonally frozen ground." was rephrased as "… even more so in cold regions considering the rapid change of seasonally frozen ground under climate warming ."

4. - L. 17/L. 80/L. 267: Abstract and manuscript discuss in several places "the ice effect" or "soil ice effect". Please clarify briefly which effect of ice is referred to when using the term, it is not always clear from the context that this refers to the impact of freezing-thawing on energy and water properties.

**Response**: L. 17: Rephrased "the ice effect" as "the impact of soil ice content on energy and water transfer properties"

L. 80: Rephrased "ice effect" as "the effect of soil ice content on hydrothermal properties"

L. 267: Rephrased "soil ice effect" as "the effect of soil ice content on hydrothermal properties"

5. - L. 38: replace "ecosystem" with "an ecosystem".

**Response**: Agree and replaced "ecosystem" with "an ecosystem"

6. - L. 82: Replace "does" with "do"

**Response**: Agree and replaced "does" with "do"

7. - L. 87: "process" is not clear here – maybe "detailed soil mass and energy transfer scheme"?

**Response**: Agree to use "detailed soil mass and energy transfer scheme"

8. - L. 90: replace "facilitating" with "facilitate"

**Response**: Replaced "facilitating" with "facilitate"

9. - L. 167: remove "however"

**Response**: Removed "however"

10. - L. 279: Is any information on longwave radiation going into the model, or is it only used for evaluation?

**Response**: The incoming longwave radiation is calculated as the function of the downward atmospheric radiation and the sky view factor. Here the measured downwelling longwave radiation was used as the input.

The outgoing longwave radiation is calculated using the surface radiative temperature, via the Stefan-Boltzmann law, in the model.

Surface energy balance (latent heat, sensible heat, ground heat and radiation fluxes), expressed as function of surface temperature, are solved iteratively to obtain the surface temperature with the constraints of the energy balance closure.

After successfully solving the surface temperature, the surface heat fluxes are recalculated with the updated surface temperature.

11. - L. 282: "second values": Does this refer to values every second, or simply any value between the two hourly values that you have data for? Please clarify.

**Response**: Here we want to say that the value at the specific second which is interpolated from two hourly values. Now we replaced "second values" with "values at every second"

12. - L. 357: Check spelling of "cryosuction"

**Response**: Now replaced "cryoscution" with "cryosuction". Thanks a lot.

13. - L. 422: I would not refer to these as "model scenarios" but "model versions"

**Response**: Agree and replaced "model scenarios" with "model versions" here.

**Referee #2:** Pierrick Lamontagne-Hallé, pierrick.lamontagne-halle@mail.mcgill.ca

Thanks a lot for the helpful comments from Pierrick Lamontagne-Hallé, we made the point-by-point response as below in blue font.

*General comments:* I would like to thank the authors for fully addressing my concerns with their responses and for applying the requested changes in their manuscript. I feel that the manuscript has greatly improved from the open discussion review process and, as a result, I think it is now suitable for publication and would be very relevant to The Cryosphere. I would also like to congratulate the authors for a well-structured study that I find personally very interesting. Below are a few more minor comments and suggestions.

1. Title: I do not think the "On" in the title is necessary.

**Response**: We deleted "On" from the title. The updated title is "The Role of Vadose Zone Physics in the Ecohydrological Response of a Tibetan Meadow to Freeze-Thaw Cycles".

2. Line 58: MarsFlo became the Advanced Terrestrial Simulator a few years ago and now incorporates way more processes than it used to. I suggest the authors to use a more up-to-date reference such as Painter et al. (2016). Other good recent examples also include the numerous cryohydrogeologic numerical models presented in Grenier et al. (2018) and Lamontagne-Hallé et al. (2020).

**Response**: We agree and rephrased this sentence as "Concurrently, researchers developed dedicated models, e.g., SHAW (Flerchinger and Saxton, 1989), HYDRUS (Hansson et al., 2004), MarsFlo (Painter, 2011) and its successor Advanced Terrestrial Simulator (Painter et al., 2016), and STEMMUS-FT (Yu et al., 2018; Yu et al., 2020), implementing the soil water and heat coupling physics for frozen soils (see for reviews of the relevant models in Kurylyk and Watanabe, 2013; Grenier et al., 2018; Lamontagne-Halle et al., 2020).".

3. Line 79: This is a personal preference, but I would argue that almost all modelers consider their models to be "state-of-the-art". Considering this, I think these words rather qualify the opinion that the authors have towards their own modelling tool and are therefore unnecessary.

**Response**: We deleted these words "state-of-the-art" from the context.

4. Lines 94-100: I do not understand why the authors move this paragraph to this section, but I think it belongs more to the introduction. They could also consider deleting it if the authors wish to reduce the length of the manuscript as I do not feel it adds any useful information.

**Response**: We agree and deleted these text.

5. Line 122: The authors talk about "A few dedicated SMST profiles", although it seems like only one profile has been used to compare to the modelling results. Can the authors either precise which profile has been used (on Figure 1) or how many profiles have been used?

**Response**: Yes. For this study, one soil moisture and soil temperature profile (i.e., the central experimental site, 33°54'59"N, 102°09'32", Figure 1) was used to validate the model simulations.

We rephrased this sentence as "The dedicated SMST profile (central station, Figure 1) …".

6. Lines 126-131: According to me, this paragraph belongs in the section 2.2.

**Response**: Agree and moved this paragraph to Section 2.2.1. We changed the heading of Section 2.2.1 into "Land Surface Energy and Carbon Fluxes and Vegetation Dynamics" accordingly.

7. Line 189-190: "the thermal effect on water flow" is rather imprecise. One could see the temperature-dependence of hydraulic conductivity as a thermal effect on a water flow, which is consider in T&C-FT. Do the authors simply mean the thermal effect on water viscosity?

**Response**: Yes. We agree and rephrased it as "the thermal effect on water viscosity". Thanks a lot.

8. Lines 206-207: Unnecessary parenthesis after e.g.

**Response**: We deleted the parenthesis here.

9. Line 307: "because the true winter precipitation is difficult to observe" is repetitive with "can be partly attributed to the uncertainties of observed winter precipitation events" written earlier in the same sentence.

**Response**: We deleted the text "because the true winter precipitation is difficult to observe".

10. Line 317: "Figure S2-3" Do the authors mean Figure S2 and Figure S3? If yes, I think this should be written "S2 and S3".

**Response**: Agree and rephrased "Figure S2-3" with "Figure S2 and S3".

11. Line 417: This sentence does not make sense to me. I recommend the authors to read it again and make sure there are no words missing.

**Response**: This sentence is to say that we should carefully corroborate these data when using it. Nevertheless, the comparison results of three models, with different vadose zone physics, were not affected.

We rephrased this sentence as "These problems clearly suggest that care should be taken to the data mutual corroboration issue, …"

12. Section 5: I think this should be a sub-section of section 4 as this should be part of the discussion.

**Response**: Section 5 was changed into Section 4.7 and Section 6 was changed into Section 5 accordingly.

13. Lines 461-463: I respectfully disagree with this sentence. I highly doubt the importance of heat advection is limited to the frozen period. Kane et al. (2001), for example, states that heat advection can be quite important during freshet due to increased infiltration from snowmelt. Other studies in permafrost environments had similar conclusions (Chen et al., 2020; Sjöberg et al., 2016).

**Response**: Sorry for the unprecise sentence and the confusing definition of the frozen period here. We focused here on the difference among models in simulating the convective heat flux.

Indeed we agree that heat advection/convection can be quite important with the increased infiltration water flow from snowmelt (Kane et al., 2001). It demonstrates that the coupling between water and heat flow is necessary for addressing the heat advection problem, which is exactly the major cause of the difference in the simulated convective heat flux between the unCPLD and CPLD models.

We rephrased this sentence as "The difference among models in simulating the liquid water flux-induced convective heat flux is mostly relevant to the freezing or thawing process".

14. Line 467-469: I am surprised by this sentence. I agree that latent heat will slow down the freezing or thawing process, but I do not think the same can be said about heat advection. Heat advection has been shown to increase thawing rate in permafrost environments (Dagenais et al.,

2020; Devoie et al., 2019). I recommend verifying this statement and specify how Figure 6 precisely demonstrates it.

**Response**: We agree and would like to highlight the role of latent heat in freezing and thawing process here.

We rephrased this sentence as "The latent heat released by freezing and consumed by the melting processes slows down the freezing/thawing process and decreases the diurnal and seasonal temperature fluctuations (Figure 7).". "Figure 6" was replaced by "Figure 7". Thanks a lot.

15. Table 1: I think including such a table is very useful. However, I think it would be easier to read if the more information was included in the table itself instead of the underlying note. For example, it seems like the table is large enough to include "Latent heat" and "Convective heat" instead of "LH" and "CH". I would also simply add "on soil properties" next to "ice effect".

**Response**: Agree and replaced "LH" with "Latent heat" "CH" with "Convective heat".

Added "on soil properties" after "ice effect". Thanks a lot.

16. Figure 1: I do not think this figure adds any useful information. The second image is particularly painful to read as labels are stacked on top of each other and the legend is very small. If the authors decide to keep this figure, I strongly recommend improving its formatting as it currently looks like a quick snapshot from ArcMap. For example, I do not think the labels are necessary as these are never discussed in the text.

**Response**: Figure 1 shows the location of Maqu Observation network and the dedicated Soil moisture and soil temperature profile (Central station) we used in this study.

We prefer to keep this figure. We deleted the unnecessary labels and highlighted the location of the Central station. The quality of this figure was enhanced.

17. Figure 2b: I think this graph looks too complicated for what it represents. Instead of referring to thawing and freezing fronts, why not simply make this graph look like a cross-section (with the same x-axis as date) where frozen and unfrozen layers are clearly represented by different colors? That would probably be easier to interpret for the readers. See sketch attached to this review for a very quickly drawn example. Furthermore, it is very unclear what the black line represents (FTFP).

**Response**: Figure 2b presents the measurements or the interpolation results from the measured temperature field (2.5 cm, 5 cm, 10 cm, 20 cm, 60 cm, and 100 cm). The propagation of freezing and thawing front (FTF) is derived from the measured temperature field. Here we did not extrapolate to the surface soil layers.

FTF consists of two kind of curves, i.e., the primary FTF and secondary FTF occurring at top soil layers. For the primary FTF, the freezing front (FF) was presented in blue color and the thawing front (TF) was presented in the red color. The secondary FTF was presented in the black color at the top soil layers.

Here we prefer to use the current form of Figure 2b. First, we want to present the 'measured' dynamics of FTF, not including the extrapolation values.

Second, the frequent freezing/thawing cycles at surface soil layers (e.g., 2.5 cm) might not be well captured by the cross-section figure.

To make it easier and clearer to the readers, we added more explanation in the legend and caption of Figure 2, clearly indicating the meaning of each curve. The updated caption of Figure 2 is "Figure 2. (a) Observed cumulative precipitation (P) and evapotranspiration (ET) and (b) observed propagation of freezing/thawing front, with the blue, red, and black color for the primary propagation of freezing front and thawing front (FF & TF), and the secondary freezing-thawing front (sFTF) occurring at top soil layers, respectively, for the period 25 Mar. 2016- 12 Aug. 2018 at Maqu site.".

18. Figure 7: I don't fully understand the choice of the temperature scale here. Why going from 50 to -25°C while the temperature data seems to be within 25 and -20°C? It makes it harder to see the differences between the different simulations.

**Response**: The original temperature scale was selected to leave some space for the upper part of the plot (5 cm), avoiding the overlaps with the plot of Precipitation.

We readjusted the temperature scale within 30 and -20°C now.

19. Figure 8: I think the direction of the water flux should be specified (e.g., positive = upward flow).

**Response**: We added the relevant text in the caption of Figure 8 as "… vertical water flux (positive value indicates upward water flow) …"

[revised manuscript text omitted]

---

## Author Response (AR3)

Comments from Editor Ylva Sjöberg:

*General comments:* Thank you for submitting your revised manuscript tc-2020-88 ('The Role of Vadose Zone Physics in the Ecohydrological Response of a Tibetan Meadow to Freeze-Thaw Cycles'). You have responded to the remaining comments from the reviewers and made minor revisions on the text based on their input. I ask you now for a few technical corrections, after which the manuscript is ready for publication in TC.

Really appreciate your dedicated time and effort in processing and reviewing our manuscript. Please find below our response in blue font.

L56: Do you mean "cold region ecosystems"?
Yes and change into "cold region ecosystems".

L86: I don't think you need "the" here (third word on row)
Agree and deleted.

L86-89: I find this sentence hard to read. Perhaps changing from "explicit considering" to "explicitly considering" fixes the sentence, but I recommend you ask a native speaker to double check the formulation.
"Explicit considering" was changed to "explicitly considering". Thanks a lot.

L93: "enables to evaluate" changed to "enables evaluation of"
Agree and changed "enables to evaluate" to "enables evaluation of"

L209: add space between ";" and "Hanson"
Agree and added space between ";" and "Hanson"

L224: Write out full name of model first, followed by abbreviation in parenthesis.
Agree to change.

L314: change from "(Figure 4)." to "in Figure 4."
"(Figure 4)." was changed to "in Figure 4."

L392: replace "/" in unCPLD/CPLD with "and" (if that is what you mean).
Yes. "unCPLD/CPLD" was changed to "unCPLD and CPLD". Thanks a lot.

L404: Same as above for "water/temperature"
"water/temperature" was changed to "water and temperature".

L480: Do you mean "not much"?
Yes. "no much" was changed to "not much"

[revised manuscript text omitted]